# RAS at the Golgi antagonizes malignant transformation through PTPRκ-mediated inhibition of ERK activation

Berta Casar[1,2], Andrew P. Badrock[3], Iñaki Jiménez[1], Imanol Arozarena[1,7], Paula Colón-Bolea[1], L. Francisco Lorenzo-Martín[2,4,5], Irene Barinaga-Rementería[3], Jorge Barriuso[3], Vincenzo Cappitelli[1], Daniel J. Donoghue[6], Xosé R. Bustelo [2,4,5], Adam Hurlstone [3] & Piero Crespo[1,2]

RAS GTPases are frequently mutated in human cancer. H- and NRAS isoforms are distributed over both plasma-membrane and endomembranes, including the Golgi complex, but how this organizational context contributes to cellular transformation is unknown. Here we show that RAS at the Golgi is selectively activated by apoptogenic stimuli and antagonizes cell survival by suppressing ERK activity through the induction of PTPRκ, which targets CRAF for dephosphorylation. Consistently, in contrast to what occurs at the plasma-membrane, RAS at the Golgi cannot induce melanoma in zebrafish. Inactivation of PTPRκ, which occurs frequently in human melanoma, often coincident with TP53 inactivation, accelerates RAS-ERK pathway-driven melanomagenesis in zebrafish. Likewise, tp53 disruption in zebrafish facilitates oncogenesis driven by RAS from the Golgi complex. Thus, RAS oncogenic potential is strictly dependent on its sublocalization, with Golgi complex-located RAS antagonizing tumor development.

[1] Instituto de Biomedicina y Biotecnología de Cantabria (IBBTEC), Consejo Superior de Investigaciones Científicas (CSIC)—Universidad de Cantabria, Santander 39011, Spain. [2] Centro de Investigación Biomédica en Red de Cáncer (CIBERONC), Instituto de Salud Carlos III, Madrid 28029, Spain. [3] Division of Cancer Studies, School of Medical Sciences, Faculty of Biology, Medicine and Health, The University of Manchester, Manchester M13 9PL, UK. [4] Centro de Investigación del Cáncer, CSIC-Universidad de Salamanca, Salamanca 37007, Spain. [5] Instituto de Biología Molecular y Celular del Cáncer, CSIC-Universidad de Salamanca, Salamanca 37007, Spain. [6] Department of Chemistry and Biochemistry, University of California, San Diego, La Jolla CA92093, USA. [7] Present address: Navarrabiomed-FMS IDISNA, Pamplona, Navarra 31008, Spain. These authors contributed equally: Berta Casar, Andrew P. Badrock. Correspondence and requests for materials should be addressed to A.H. (email: Adam.Hurlstone@manchester.ac.uk) or to P.C. (email: crespop@unican.es)

Signals conveyed through RAS family GTPases play critical roles in multiple biochemical processes, hence in key biological decisions at the proliferation-differentiation-survival crossroads. Their importance in cell physiology is highlighted by the dramatic results of their malfunction, *RAS* mutational activation being detected in about 30% of human tumors[1]. It has long been known that in order to be functional RAS proteins must associate with the plasma-membrane[2]. However, a wealth of data accumulated over the past decades has firmly established that, in addition to the plasma-membrane, RAS is also present and functional at endomembranes, such as the endoplamic reticulum (ER), endosomes, and the Golgi Complex (GC)[3]. This has led to the initial concept of a single source of RAS signals, now being envisioned as the integration of subcellular location-specified subsignals, with output variability depending on the availability of regulatory and effector molecules at the different platforms from which RAS signals emanate[4,5].

While it is firmly established that at its diverse sublocalizations: RAS is subject to site-specific regulation by different exchange factors[6–8]; engages different effector pathways[9,10]; and switches on distinct genetic programs[11], the participation of each of the RAS signaling platforms in defined RAS-mediated biological outcomes, remains unclear. Such is the case for carcinogenesis: how RAS sublocalization impacts on its potential to drive malignancy remains an open question.

This uncertainty is particularly relevant in the case of endomembranes. While there is little doubt about the participation of RAS signals generated at plasma-membrane microdomains in carcinogenic processes[12], the involvement of RAS signals coming from endomembranes, particularly the GC, remains obscure. It is known that pools of H- and NRAS, but not of the most oncogenic isoform KRAS, reside at the GC and therein they can productively engage downstream effectors[9,13–17]. However, the association of the RAS GC pool to malignancy is understudied and the available data is solely restricted to cell culture approaches that have yielded inconclusive results[9,15,18]. Herein, we have aimed at filling this gap by using diverse cellular and animal models for studying the role played by RAS signals emanating from the GC in cancer and demonstrate that RAS at this organelle antagonizes tumor development

## Results

**RAS at the GC is distinctively activated by physiological stimuli.** To gain an initial insight into the participation of the RAS GC pool in key processes relevant to carcinogenesis, such as proliferation, differentiation, and survival/apoptosis, we investigated whether agonists that yield such effects physiologically could activate RAS at the GC. To this end, we utilized MCF-7 cells .This mammary epithelial cell line undergoes different fates depending on the agonist: EGF induces proliferation whereas heregulin (HRG) evokes adipocytic-like differentiation[19]. To monitor RAS activation specifically at the GC, we used as a probe a construct expressing wild-type HRAS N-terminally fused to the KDEL receptor harboring the mutation N193D, which prevents it from redistributing to the ER, making it a permanent GC resident[20]. This probe has been successfully utilized in our previous studies[8,9,21]. It was found that neither EGF nor HRG-induced GDP/GTP exchange on RAS at the GC, which did however respond to the presence of overexpressed RASGRP1[7] (Fig. 1a). This finding was further substantiated by analyses in live cells, using RAF RBD fused to three eGFPs in tandem to indicate the presence of RAS-GTP[22]. When this construct was co-expressed with cherry-HRAS in cells stimulated with EGF, RAS activation was exclusively detected at the cellular periphery, but not in internal structures (Fig. 1b and Supplementary Figure 1B).

Identical results were obtained under HRG stimulation (Supplementary Figure 1C).

Considering that treatment of MCF-7 cells with TGF-β provokes apoptosis[23], we tested whether such a stimulus would induce RAS activation at the GC. This was the case, though, interestingly, TGF-β-evoked RAS GDP/GTP exchange at the GC was not accompanied by ERK activation (Fig. 2a). When analyzing RAS activation in response to TGF-β in live cells, it could be noticed that, unlike EGF and HRG, this ligand evoked a prominent RAS activation at the nuclear periphery, consistent with GC localization (Fig. 2b and Supplementary Fig 1A, D and E). These data indicate that while physiological ligands that yield proliferative or differentiation responses fail to evoke RAS GDP/GTP exchange at the GC, apoptogenic stimuli induce RAS activation at this organelle.

**RAS activation at the GC induces apoptosis and prevents cellular transformation.** In light of our data showing that proapoptotic stimulation evokes activity of the RAS pool at the GC, we next tested whether RAS activation at this organelle triggered apoptosis by itself. For this purpose, we specifically directed HRASV12 constitutively active mutant to the GC using the KDELr tag (KDELr-HV12 hereafter). Remarkably, expression of this construct, but not KDELr alone, in MCF-7 cells was sufficient to induce apoptosis, comparable to that elicited by TGF-β (Fig. 3a and Supplementary Figure 2A).

H- and NRAS traffic between the GC and the plasma-membrane depending on their acylation status[24]. The drug palmostatin B (palm B) prevents RAS cycling between both compartments and fosters RAS accumulation at endomembranes by suppressing its deacylation[25]. Thus, we reasoned that trapping active RAS at the GC using palm B should elicit an apoptogenic response. Indeed, in MCF-7 cells palm B treatment induced a mild apoptotic effect that was markedly augmented in cells expressing HRASV12 (Fig. 3b and Supplementary Figure 2B). Likewise, siRNA-mediated knockdown of palm B target Acyl protein thioesterase 1 (APT-1) also elicited a potent apoptotic response (Fig. 3c).

To further substantiate this hypothesis, we evaluated apoptosis following palm B treatment in a series of tumor cell lines harboring different oncogenes. Noticeably, palm B resulted in a marked accumulation of RAS at the GC (Supplementary Figure 3) and readily induced cell demise in cell lines with mutant H- and NRAS, but not in those expressing BRAF or KRAS, that does not traffic through the GC[14] (Fig. 3d). The protein palmitoylation inhibitor 2-bromopalmitate that evokes NRAS and HRAS accumulation in the GC[26] yielded similar results (Supplementary Fig. 4a). The possibility existed that the apoptotic response following palm B treatment was caused by the depletion of the RAS plasma-membrane pool as a consequence of the disruption of its acylation cycle, rather than due to the accumulation of RAS at the GC. To rule this out, we cultured tumor cell lines at 21 °C. At this temperature, post-GC transport is stopped[27] blocking the transit of newly-synthesized RAS, in the absence of any short-term alteration of the RAS pool already at the plasma-membrane (Fig. 3e and Supplementary Figure 3). Similarly to palm B treatment, culture at 21 °C potently induced apoptosis in cell lines expressing mutant H- and NRAS, but not in those harboring BRAF or KRAS (Fig. 3f), indicating that accumulation of mutant RAS at GC was responsible for triggering the observed apoptogenic effects. To validate further this point, we used "Rasless" fibroblasts. In these, the complete absence of Ras isoforms is not sufficient to trigger apoptosis[28], but it was significantly induced by transfecting GC-targeted, but not plasma-membrane-targeted, HRASV12 (Supplementary Figure 4B). While a potent

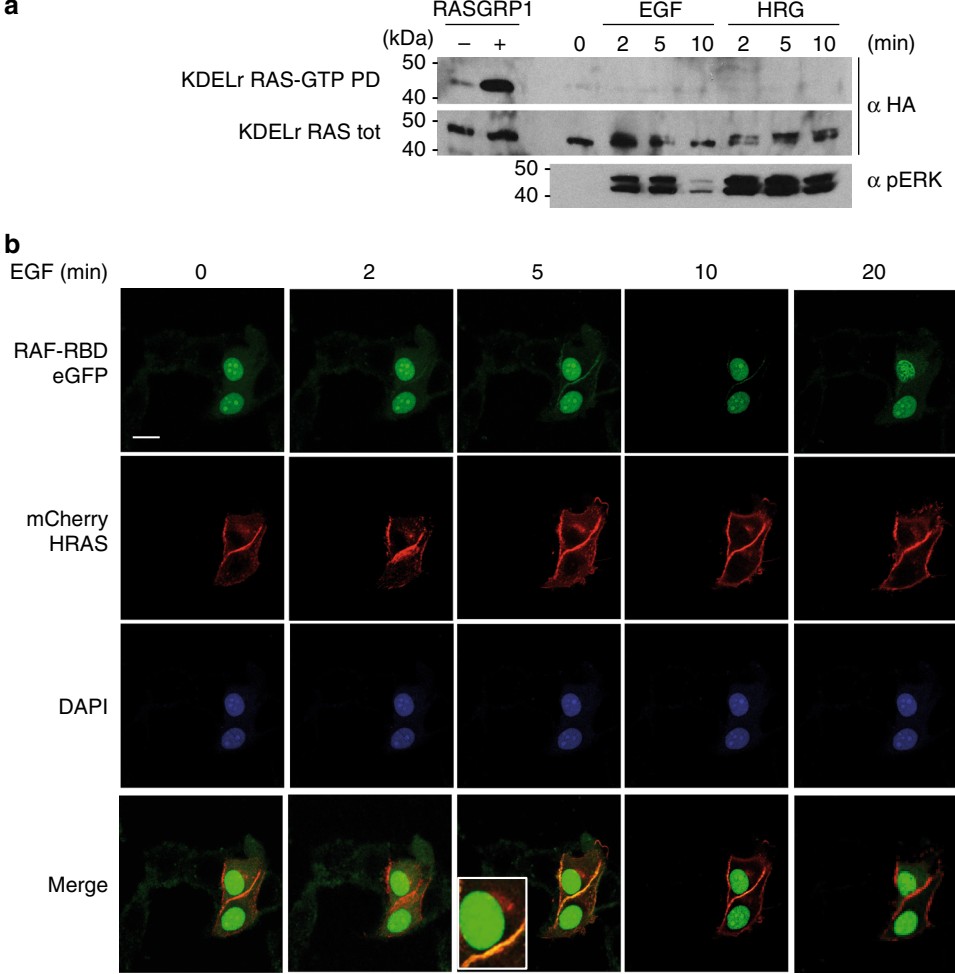

**Fig. 1** GC RAS response to proliferative stimuli. **a** RAS is not activated at the GC by proliferative stimuli. MCF-7 cells transfected with HA-tagged KDELr HRAS (0.5 μg), unstimulated (0) or treated with EGF (50 ng/ml) or HRG (30 ng/ml) for the indicated times. Cells transfected with RASGRP1 (1 μg) (+) serve as positive control. GTP loading was assayed by GST-RBD pull-down (RAS-GTP-PD). **b** RAS activation by EGF is restricted to the plasma-membrane. MCF-7 cells transfected with constructs expressing cherry-HRAS and the RAS-GTP biosensor E3-R3 (RAF-RBD) (1 μg each) and stimulated for the indicated times. Scale bar = 10 μm

inductor of apoptosis, HRASV12 at the GC did not evoke senescence or cell cycle arrest (Supplementary Figure 4C, D).

Given its apoptogenic activity, we evaluated whether the presence of activated RAS at the GC would prevent cellular transformation. For this purpose, we assayed whether KDELr-HV12 could antagonize the transformation of NIH3T3 fibroblast as induced by bona fide oncogenes. Indeed, co-transfection of the aforementioned construct was sufficient to significantly diminish the number of transformed foci generated by potent oncogenes, such as KRAS, NRAS, and HRAS, irrespective of the subcellular localization from which its signals emanated, as well as other types of oncogenes, such as v-Src, v-Sis, and ERB2 (Supplementary Table 1). Overall, these sets of data demonstrate that the presence of activated RAS, both endogenous and ectopic, at the GC is sufficient to stimulate an apoptotic response and forestall malignant transformation.

As the GEF RASGRP1 has been identified as a GC RAS activator[6,7], we analyzed its role in GC RAS-mediated apoptotic response. Indeed, ectopic expression of RASGRP1 in MCF-7 cells markedly induced apoptosis (Fig. 4a). $Ca^{2+}$ positively regulates RAS at the GC[6] via RASGRP1 phosphorylation at serine 332[29]. In agreement, a RASGRP1 S332D phospho-mimetic mutant readily evoked cell death, whereas a S332A phosphorylation-defective

mutant failed in this task (Fig. 4a). However, RASGRP1 expression levels in MCF-7 cells are minimal (Fig. 4b). Accordingly, TGF-β-evoked GC RAS nucleotide exchange (Fig. 4c) and apoptosis (Fig. 4d) were unaffected by a siRNA against RASGRP1, suggesting that other yet unidentified GEFs must be responsible for TGF-β-induced RAS activation at the GC in these cells.

**RAS at the GC downregulates ERK activation.** It was important to elucidate the mechanism whereby RAS activation at the GC elicits cell death. Deficient activation or blockade of the RAS effector pathway leading to the activation of the mitogen-activated protein kinases ERK1 and 2 (ERK hereafter) underlies many apoptotic processes[30]. Previous studies from our lab have shown that RAS at the GC fails to stimulate substantial ERK activity levels[9,21]. Indeed, as shown in Fig. 2a, TGF-β-induced RAS activation at the GC did not provoke any detectable ERK phosphorylation. More interestingly, RAS constitutive activity at the GC resulting from the expression of KDELr-HV12 markedly interfered with ERK phosphorylation as induced by an external agonist like EGF or by a potent oncogene like HRASV12, but not by a constitutively active form of MEK1, MEK E (Fig. 5a).

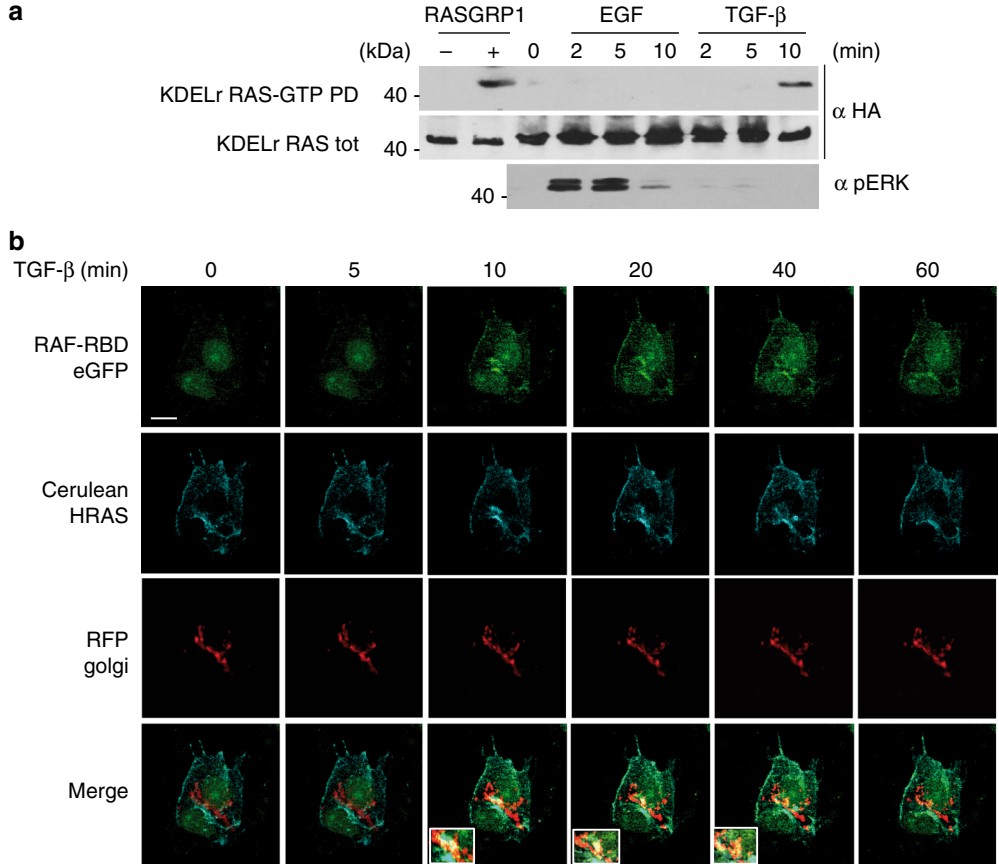

**Fig. 2** GC RAS response to apoptogenic stimuli. **a** Apoptogenic stimuli activate RAS at the GC. MCF-7 cells transfected with HA-tagged KDELr HRAS (0.5 µg), unstimulated (0) or treated with TGF-β (5 ng/ml) for the indicated times. Cells transfected with RASGRP1 (1 µg) (+) serve as positive control. GTP loading was assayed by GST-RBD pull-down (RAS-GTP-PD). **b** RAS activation in endomembranes induced by TGF-β. MCF-7 cells transfected with constructs expressing cerulean-HRAS and the RAS-GTP biosensor E3-R3 (RAF-RBD) (1 µg each) and stimulated for the indicated times. GC was revealed by the RFP Golgi probe. Insets show areas of prominent RAS-GTP accumulation. Scale bar = 10 µm. See also Supplementary Fig. 1

Noticeably, GC RAS inhibitory effect was restricted to the ERK pathway, as it did not alter the activity of other routes such as the PI-3K pathway (Supplementary Figure 5A). To eliminate the possibility that this was an artifact resulting from the expression of the chimeric construct KDELr-HV12, we tested whether a similar response could be obtained by stimulating the endogenous RAS GC pool. This was accomplished by using the KDELr tag to tether to the GC the CDC25 domain of RASGRF1 (KDELr-CDC25)[21,31], thereby achieving a potent activation of RAS specifically at this organelle (Supplementary Figure 5B). By using this construct it was found that activation of the endogenous RAS GC pool evoked a suppressive effect on ERK activation, identical to that one resulting from the ectopic expression of KDELr-HV12 (Fig. 5b).

Since the GC is made up of two functionally and structurally different networks: the *cis* Golgi network (CGN) and the *trans* Golgi network (TGN), it was of interest to determine from which of these compartments RAS would be exerting its suppressive role on ERK activation. Since the KDELr is a CGN anchor[20], we engineered a construct to specifically deliver HRASV12 to the cytosolic leaflet of the TGN. This was achieved by the fusion of SCG-10 34 N-terminal amino acids[32] (Supplementary Figure 5C). Noticeably, while KDELr-HV12 markedly inhibited ERK phosphorylation and kinase activity induced by the oncogene HRASV12, SCG10-HV12 failed to do so (Supplementary Figure 5D), indicating that GC RAS inhibitory effect on ERK emanates specifically from the CGN.

We then sought to identify the effector pathway(s) that GC RAS utilizes to downregulate ERK activation. For this purpose, we added the KDELr tether to a series of HRASV12 switch-II domain mutants known to specifically activate defined effector routes[33]. It was found that the G37 mutant, that specifically activates RAL GEFs, but not those that exclusively signal through CRAF (S35) or PI-3K (C40), could downregulate ERK activation (Fig. 5c). To further substantiate this point, we also analyzed the suppression of ERK activation by other bona fide RAS effectors, uncharacterized with respect to their response to mutations in the RAS switch-II domain. While HRASV12-induced ERK phosphorylation was markedly attenuated by overexpression of RALGDS, other RAS effectors failed to do so (Fig. 5d), pointing to the RAL GEFs effector pathway as the only one responsible for antagonizing ERK activation. In this respect, even though overexpression of RALGDS was not sufficient to stimulate an apoptotic response in MCF-7 cells, it synergized with KDELr-HV12 to induce cell death as potently as the ERK pathway inhibitor U0126 (Fig. 5e). In summary, these results demonstrate that RAS signals emanating from the GC exert a negative impact on ERK activation, as stimulated by external agonists or by internal cues such as oncogenic RAS signals coming from other subcellular locations.

**PTPRκ induces apoptosis by inhibiting ERK activation.** It was essential to unravel the mechanism whereby ERK activation was inhibited by RAS signals coming from the GC. RAS activation at

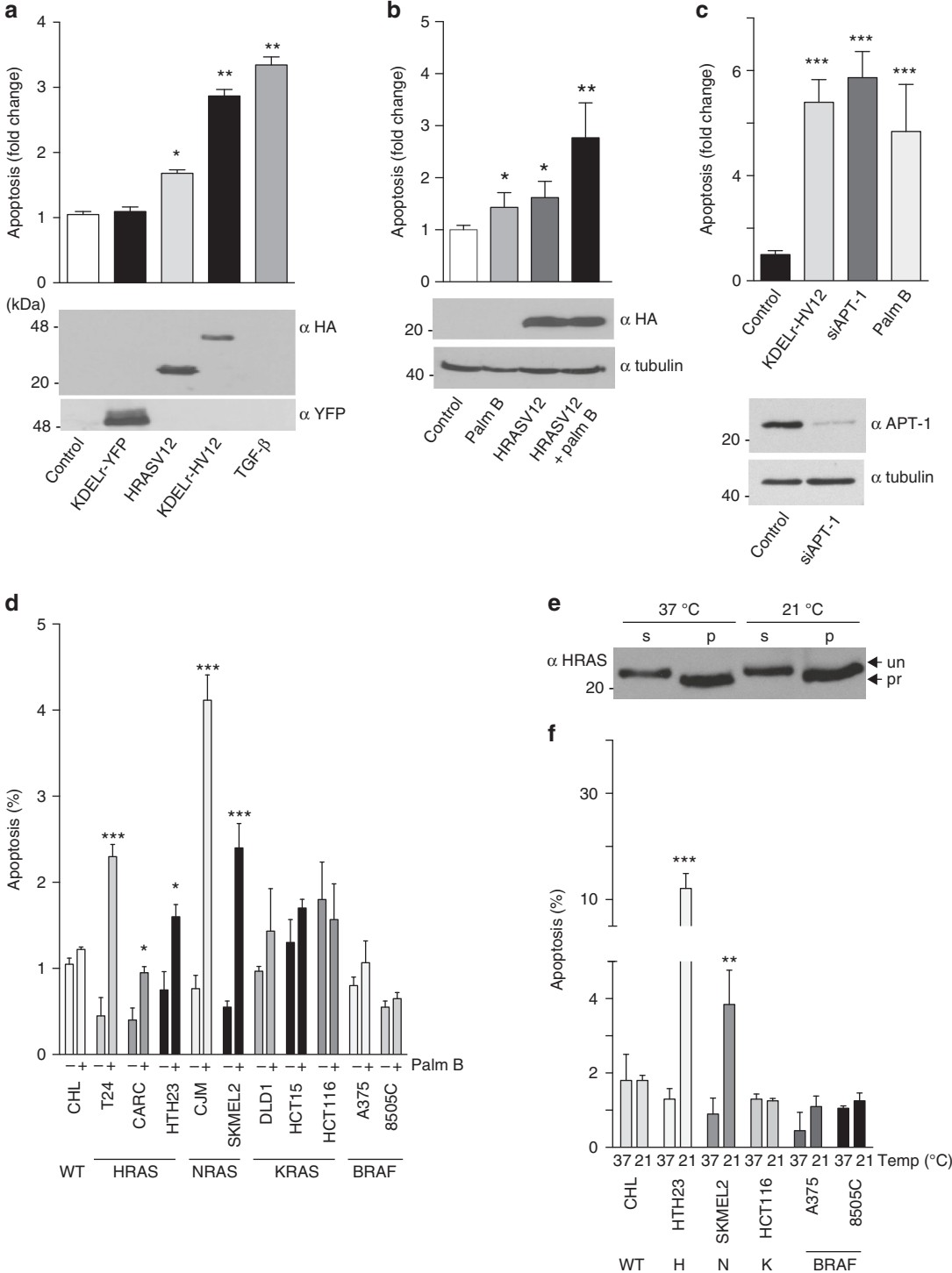

**Fig. 3** Induction of apoptosis by RAS activation at the GC. **a** Induction of apoptosis in MCF-7 cells transfected with the indicated constructs (1 μg each) or treated with TGF-β (5 ng/ml for 12 h). Apoptosis was evaluated by annexin V detection using the Guava/nexin assay. **b** Apoptosis in response to palmostatin B treatment (10 μM, 24 h) in cells transfected with the indicated constructs (1 μg each). **c** Apoptosis in response to APT-1 knockdown using a siRNA (100 nM). Lower panel: expression levels of APT-1 in control and siRNA-treated cells. **d** Apoptotic response to palmostatin B of tumor cells harboring the indicated oncogenes. WT = wild-type for RAS and BRAF. **e** Effects of temperature-dependent GC traffic blockade on RAS levels. RAS levels at the soluble (S) and particulate (P) fractions of SKMEL2 cells cultured at the indicated temperatures. Arrows (un/pro) indicate processed and unprocessed RAS forms. **f** Apoptotic response of tumor cells harboring different oncogenes to temperature-dependent GC blockade. Cells were kept at the indicated temperatures for 24 h. In all cases data show average ± SEM from 3 (A,B,C) or 5 (D,F) independent experiments. ns >0.05; *$p < 0.05$; **$p < 0.01$; ***$p < 0.005$ by Student's $t$-test. See also Supplementary Figs. 2–4

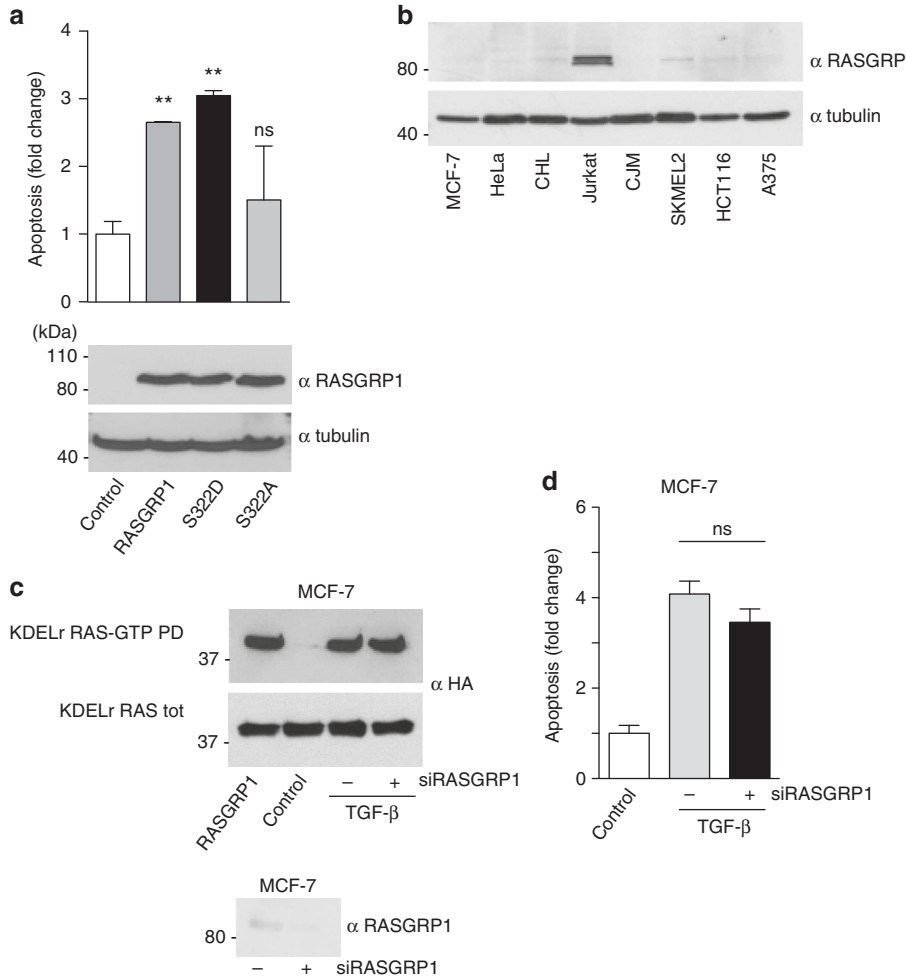

**Fig. 4** Role of RASGRP1 in the GC RAS-mediated apoptotic response. **a** Induction of apoptosis by GC-specific RAS GEFs in MCF-7 cells transfected with the indicated constructs (1 μg each). **b** RASGRP1 expression levels in the indicated cell lines. **c** RASGRP-1 role in TGF-β-induced RAS activation. MCF-7 cells transfected with HA-tagged KDELr HRAS (0.5 μg), in control cells or treated with TGF-β (5 ng/ml) for 10 min in the presence (+) or absence of a siRNA (100 nM) against RASGRP1. Cells transfected with RASGRP1 (1 μg) (+) serve as positive control. GTP loading was assayed by GST-RBD pull-down (RAS-GTP-PD). **d** RASGRP-1 role in TGF-β-induced apoptosis in MCF-7 cells. Data show average ± SEM from three independent experiments. ns >0.05; **p < 0.01 by Student's t-test

the GC did not prevent RAS access to other membrane systems (Supplementary Fig. 5A), suggesting that the inhibitory activity of KDELr-HV12 on ERK has to take place via alternative mechanisms. In a previous microarray-based study, we found that RAS at the GC could induce a limited, but highly specific subset of genes[11]. Interestingly, this included Protein Tyrosine Phosphatase receptor kappa (PTPRκ), a protein involved in TGF-β antiproliferative effects[34]. This suggested that this phosphatase could be one of the elements involved in the proapoptotic pathway of CG-localized RAS. In favor of this idea, we observed that in MCF-7 cells expressing oncogenic HRAS, TGF-β-induced suppression of ERK activation was accompanied by an upregulation of PTPRκ levels (Fig. 6a). Similarly, when we tested the ability of KDELr-HV12 to suppress ERK activation in different cell lines, this only occurred in those cells where constitutive GC RAS activity induced PTPRκ expression (Fig. 6b). Further, when MCF-7 cells were co-transfected with KDELr-HV12 and increasing concentrations of an shRNA against PTPRκ, it was found that progressive downregulation of PTPRκ expression was accompanied by augmented ERK phosphorylation levels (Fig. 6c). In agreement, in HeLa cells, in which PTPRκ is not expressed (Fig. 6b), KDEL-HV12 stimulated ERK phosphorylation (Fig. 6d).

To rule out that the observed effects were due to some artefact evoked by the GC tethering cue, we utilized an alternative GC anchor: the avian infectious bronchitis virus E1 protein, containing a cis-GC targeting signal[18] (Supplementary Figure 6B, C). When targeted to the GC by this means activated HRAS proved as efficient as KDELr-tethered for inducing PTPRκ expression, downregulation of ERK phosphorylation and inducing apoptosis (Fig. 6e).

To identify the point of the RAS-ERK pathway subject to PTPRκ regulation, we analyzed the phosphorylation of the different tiers of the cascade in response to the upregulation of the phosphatase. It was found that increasing the expression of PTPRκ markedly reduced RAS-stimulated CRAF tyrosine phosphorylation and, consequently, the phosphorylation levels of MEK and ERK (Fig. 6f). Contrarily, increments on PTPRκ levels did not alter ERK phosphorylation as induced by MEK E (Supplementary Figure 6D). These results indicate that PTPRκ is probably involved in the inactivation of CRAF. Protein tyrosine phosphatase receptors bind to their substrates stably enough to be detected by co-immunoprecipitation[35]. Indeed, epitope-tagged forms of PTPRκ and CRAF readily co-immunoprecipitated when expressed in MCF-7, accompanied by a drop in CRAF tyrosine

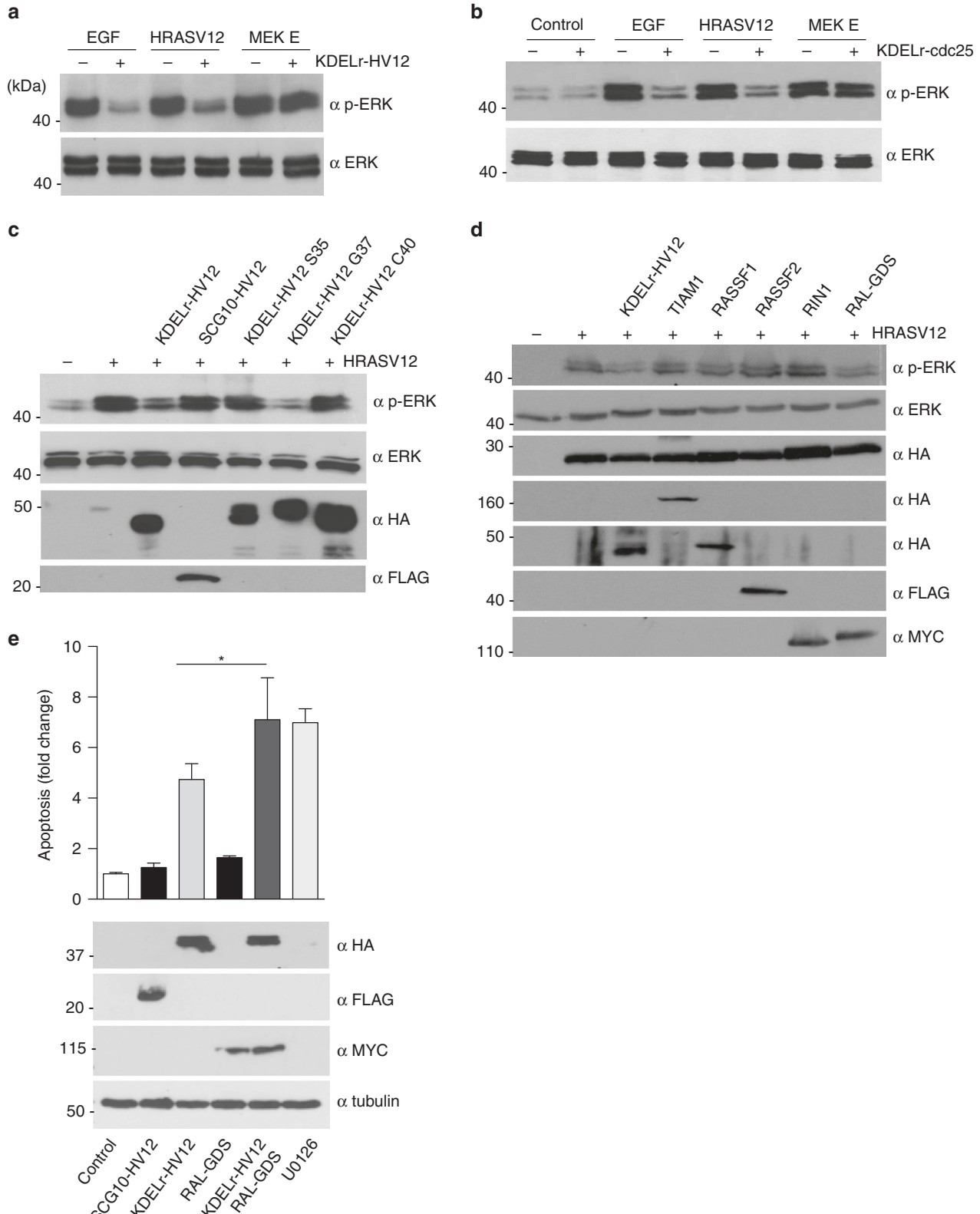

**Fig. 5** Effects on ERK phosphorylation of RAS activation at the GC. **a** MCF-7 cells stimulated with EGF (50 ng/ml) or co-transfected with constructs expressing the indicated proteins (1 μg) with (+) or without (−) KDELr-HV12 (1 μg). ERK phosphorylation was assayed by immunoblotting. **b** As in **a**, but cells were co-transfected with KDELr-Cdc25. **c** ERK phosphorylation is inhibited by GC RAS via RALGDS. Cells were transfected with HRASV12 (+) together with the indicated KDELr-tethered RAS effector domain mutants (1 μg each). **d** As in **c**, cells were transfected with constructs encoding for the indicated effector proteins (1 μg each). Representative results are shown in every case. **e** RALGDS cooperates with GC RAS for inducing apoptosis. Apoptosis in MCF-7 cells transfected with the indicated constructs (1 μg each) or treated with U0126 (10 μM, 24 h). Data show average ± SEM from three independent experiments *$p < 0.05$ by Student's $t$-test. See also Supplementary Fig. 5

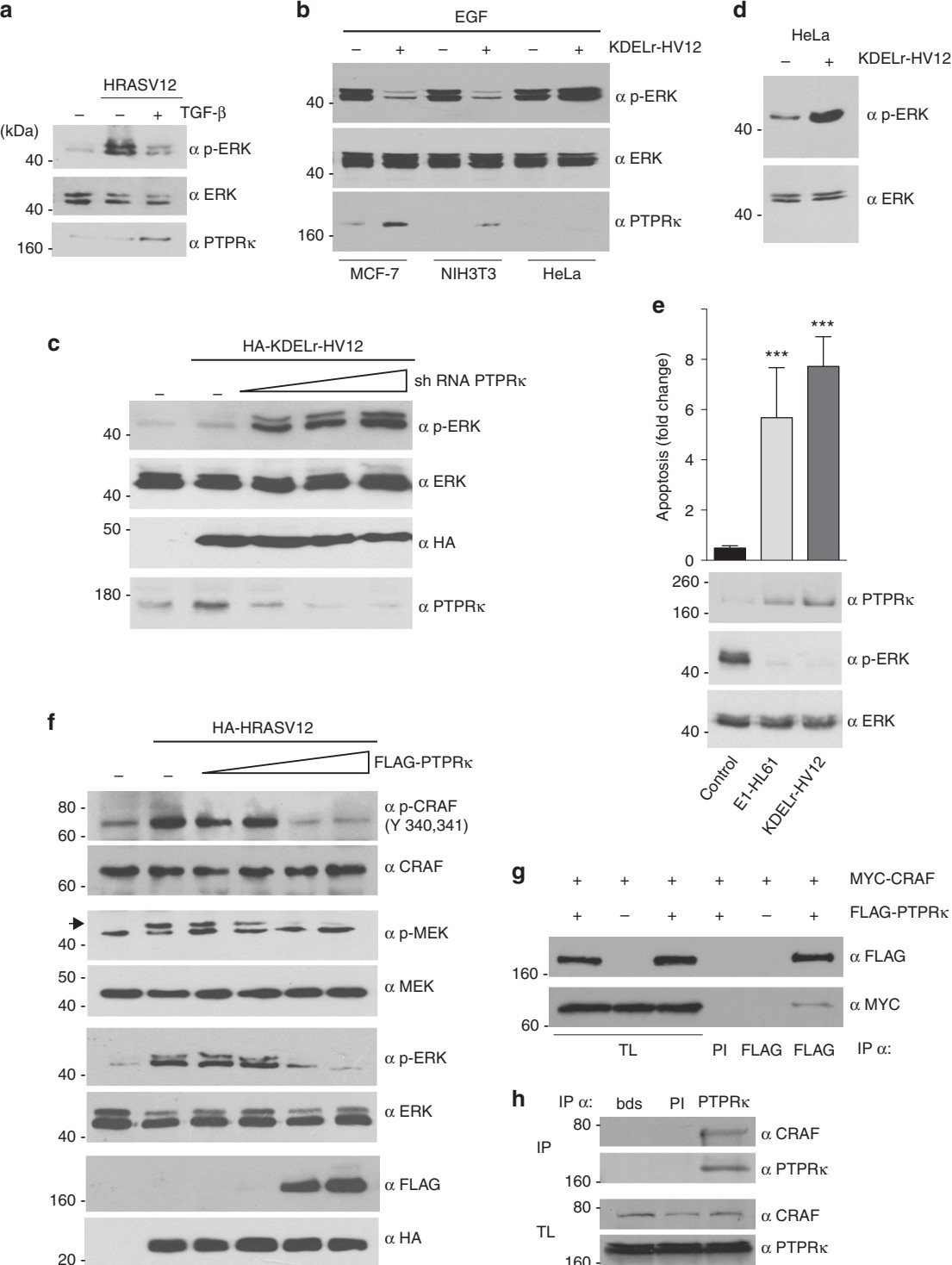

**Fig. 6** GC RAS inhibits ERK activation via PTPRκ. **a** Induction of PTPRκ expression and downregulation of ERK phosphorylation by TGF-β (5 ng/ml for 2 h) in MCF-7 cells transfected with HRASV12 (1 μg). **b** As in **a**, but by transfection of KDELr-HV12 (1 μg) in the indicated cell lines stimulated with EGF (50 ng/ml). **c** Effects of PTPRκ downregulation on ERK activation induced by GC RAS. Cells were transfected with KDEL-HV12 where shown, together with increasing concentrations (0.2–5 μg) of a construct expressing an shRNA against PTPRκ. **d** ERK activation induced by GC Ras in HeLa cells. **e** Effects on PTPRκ downregulation, ERK activation and apoptosis of GC RAS tethered by E1 and KDELr anchors (1 μg). Data show average ± SEM from three independent experiments. ***$p < 0.005$ by Student's $t$-test. **f** Effects of PTPRκ on the different tiers of the ERK cascade. Cells were transfected with HRASV12 were shown, together with increasing concentrations (0.2–5 μg) of a construct encoding for PTPRκ. **g** Co-immunoprecipitation of PTPRκ with CRAF. MCF-7 cells were transfected with the indicated constructs (1 μg) were shown (+). Immunoprecipitations were performed using pre-immune serum (PI) or anti-FLAG antibodies. TL total lysates. **h** Co-immunoprecipitation of endogenous PTPRκ and CRAF in MCF-7 cells. Control immunoprecipitations were done with pre-immune serum (PI) or beads only (bds). See also Supplementary Figure 6

phosphorylation, (Fig. 6g). Interaction was also observed for the endogenous proteins (Fig. 6h). These data demonstrate that PTPRκ quenches ERK activation by binding to and dephosphorylating CRAF.

We then tested whether PTPRκ was involved in the unleashing of apoptosis by GC RAS signals. It was found that overexpression of PTPRκ and of KDELr-HV12 in MCF-7 cells-induced apoptosis to similar extents. Likewise, we found that the siRNA-mediated knockdown of *PTPRK* ameliorated the apoptotic response triggered by KDELr-HV12 in those cells (Fig. 7a and Supplementary Figure 6E). Since PTPRκ can inhibit several signaling pathways[36], it was important to determine the extent to which the apoptotic response induced by PTPRκ expression was a consequence of its impact on the ERK cascade through inhibiting CRAF. It was found that cell death induced by either PTPRκ or KDELr-HV12 could be prevented by the co-expression of MEK1E, known to yield a constitutive ERK activation, or by BRAF that activates ERK mostly independently of CRAF (Fig. 7b). This demonstrates that suppression of ERK activation via CRAF dephosphorylation is the key event for PTPRκ-induced apoptosis.

**RAS at the GC fails to induce melanoma.** From the above data, we would predict that RAS activation at the GC is incompatible with tumourigenesis. To test this notion in an animal cancer model, we utilized melanoma induction in zebrafish. Previously, we have shown that melanocyte-directed expression of HRASV12 initially induces flat dysplastic lesions that spontaneously progress to raised, invasive melanoma[37]. We injected single-cell zebrafish embryos with constructs under the control of the micropthalmia-induced transcription factor (mitf) promoter, confirmed to overexpress site-directed RAS proteins (Fig. 8a), and monitored melanomagenesis through several weeks post-injection. To normalize for integration frequency between different constructs, we utilized *mitfa* mutants (also known as *nacre*) embryos and assessed transgene integration through rescue of pigmentation by a *mitfa* minigene also present in the construct[38]. As such, we tested the capability of HRASV12 to induce melanoma depending on its sublocalization. We observed that HRASV12 targeted to either lipid rafts (LCK-HV12) or disordered membrane (CD8-HV12)[9] could induce melanoma as efficiently as untagged HRASV12. Remarkably, none of the animals expressing KDELr-HV12 developed malignant tumors (Fig. 8b, c); moreover stripes appeared largely normal in these animals (insets, Fig. 8c). Similarly, fish expressing trans-GC targeted SCG10-HV12 did not show signs of malignancy (Supplementary Figure 7), suggesting that in zebrafish the GC RAS pool is deficient for inducing tumourigenesis.

Remarkably, at early embryological stages fish displayed similar melanocyte numbers irrespective of the site where transgenic HRASV12 was targeted (Fig. 8d). This suggested that, unlike what was previously observed in mammalian cells, active GC RAS did not induce apoptosis in zebrafish melanocytes. To test whether this was applicable to melanocytes in general, we expressed the HRASV12 site-targeted constructs in CHL cells, a human melanocytic cell line. However, in this case, and in agreement with our previous data, we observed that the presence of oncogenic HRAS at the GC, but not at other sublocalizations, unleashed a potent apoptotic response (Fig. 8e), indicating a divergence with respect to the role of the GC RAS pool along the evolutionary scale.

Overall, these findings demonstrate that RAS potential to drive carcinogenesis is strictly dependent on its subcellular localization and that mutant RAS at the GC is unable to foster melanoma, either as a result of its suppressive effect on cellular viability, as in human melanocytes, or as a consequence of its lack of effect on cellular viability/proliferation, as in zebrafish melanocytes.

**Absence of PTPRκ potentiates RAS-ERK oncogenic signals.** As the ERK pathway is a key driver of melanoma, we predicted that *PTPRK* expression and/or activity could modify disease course, especially in instances where RAS is deregulated. Using the Oncomine® platform, we found that *PTPRK* expression is significantly downregulated in melanoma compared to either benign nevi or normal skin[39] (Fig. 9a). Intriguingly, we also observed that the expression of *PTPRK* is inversely correlated with expression of LYPLA1/APT1 (Supplementary Figure 8A), which is significantly upregulated in melanoma[39] (Supplementary Figure 8B), suggesting that deregulation of the N/HRAS palmitoylation cycle in melanoma cells could be responsible for reduced *PTPRK* expression. Further supporting the association between *PTPRK* expression and melanoma progression, we found that lower abundance of *PTPRK* mRNA in tumor samples was associated with lower survival rates of melanoma patients (Fig. 9b).

We next wished to determine *PTPRK* potential as a melanoma suppressor functionally. The zebrafish genome encodes a single highly conserved *ptprk* orthologue (ENSDARG00000063416). CRISPR/Cas9 technology was utilized to generate lineages harboring either a 5 or 7 bp deletion in exon 6 of the *ptprk* gene, truncating the protein in the ectodomain of the receptor (Supplementary Figure 8C, D). Homozygous mutant animals, with *ptprk* messenger downregulated ~50-fold 6 h post fertilization (Supplementary. Figure 8E), were viable and fertile with no obvious phenotypic effect. First, we tested whether *ptprk* loss would augment GC RAS oncogenic potential. Intriguingly, in a *ptprk*-mutant homozygous background only in a few cases did KDELr-HV12 induce the formation of naevi or melanoma (Fig. 9c), suggesting that while removal of *ptprk* altered cellular responses to oncogenic RAS at the GC this was generally insufficient to yield full-blown malignancy. However, melanoma formation was significantly accelerated in homozygous *ptprk* mutant zebrafish injected with a bona fide oncogene such as NRAS G12D, compared to wild-type animals (Fig. 9d, e). Conversely *ptprk* loss did not augment melanoma induction by BRAFV600E (Supplementary Figure 8F), demonstrating that the absence of *ptprk* does not affect signalling downstream of CRAF. Overall, our findings from zebrafish models, therefore, support the proposal that *PTPRK* is a modifier of deregulated RAS driven melanoma induction.

**Downregulation of p53 fosters GC RAS oncogenic signals.** Significant mutation of PTPRκ in melanoma was previously noted: 17 different substitutions constituting 19.7% of sun-exposed melanomas[40]. Using cBioportal to examine melanoma genomes, revealed PTPRκ homozygous deletions in 3% of cutaneous melanoma samples as well as missense and non-sense mutations in 14% of cases (Fig. 10a). Mutations are distributed throughout the coding sequence and may disrupt homophilic interactions, heterotypic interactions, protein stability, or phosphatase activity[41]. Interestingly, these alterations coincide with *TP53* mutant alleles more often than expected by chance (log odds ratio = 1.050 $p < 0.001$) (Fig. 10a). These results are consistent with a specific tumor suppression function of TP53 in melanoma cases bearing mutations in specific RAS signalling elements. Further supporting this idea, while the low abundance of *PTPRK* mRNA in tumor samples is associated with lower survival rates of melanoma patients (Fig. 9b), a similar correlation was also detected in the case of patients bearing mutant but not wild-type versions of *TP53* (Fig. 10b).

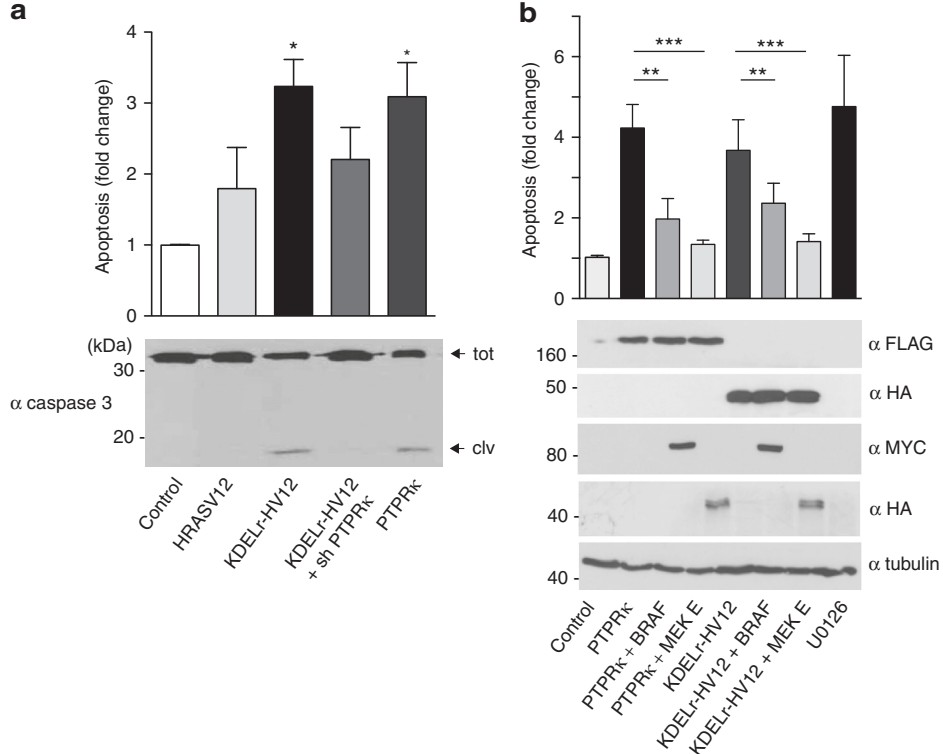

**Fig. 7** Induction of apoptosis by oncogenic RAS at the GC. **a** GC RAS-induced apoptosis dependence on PTPRκ. Apoptosis, determined by annexin V levels (bar chart) and by western blotting for activated Caspase-3, in MCF-7 cells transfected with the indicated constructs (1 μg) and siRNAs (10 nM). Arrows show total (tot) and cleaved (clv) caspase-3. **b** GC RAS- and PTPRκ-induced apoptosis require ERK downregulation, as revealed in MCF-7 cells transfected with the indicated constructs. **a**, **b** Data show average ± SEM from three independent experiments *$p < 0.05$; **$p < 0.001$; ***$p < 0.005$ by Student's $t$-test

Following these clinical data, it was important to determine whether TP53 deficiency impacted on the regulation of ERK signals and cellular survival by GC RAS that we had previously unveiled. In MCF-7 cells, in which we had shown that KDELr-HV12 repressed ERK activation and induced apoptosis (Fig. 5), inactivation of endogenous TP53 by the transfection of papillomavirus E6 protein[42] restored ERK activation levels and prevented KDELr-HV12-induced apoptosis (Fig. 10c). To further substantiate this point, we utilized *tp53*-null MEFs. In these, transfection of KDELr-HV12 failed to suppress ERK activation and to promote apoptosis, contrarily to what was observed in *tp53* wild-type cells (Fig. 10d).

Subsequently, we analysed whether the absence of TP53 reinstated the ability of GC HRAS to induce melanomas. Indeed injection of the KDELr-HV12 transgene into *tp53*$^{-/-}$ nacre zygotes resulted in the appearance of tumors after 14 weeks, as opposed to fish harboring a GFP transgene that never developed neoplasia (Fig. 10e and Supplementary Fig. 9). Overall, these results demonstrate that *TP53* status is a critical factor for the determination of the biological outcome of RAS signals emanating from the GC.

**Discussion**
More than 30 years on from their discovery, our understanding of RAS signaling and its contribution to tumorigenesis is still incomplete. An issue that still lingers in this field is how RAS sublocalization affects its oncogenic potential. Our findings add a new twist to this mystery as we reveal that RAS at the GC, which includes H- and N- isoforms but not KRAS, actually antagonize tumor formation through induction of *PTPRκ* that downregulates ERK activity, the engine at the heart of RAS transformation.

It is known that RAS oncoproteins can induce apoptosis[1]. Herein, we introduce subcellular location as a determinant of its apoptogenic potential: RAS signals emanating from the GC, and specifically from the cis-Golgi, can induce a potent apoptotic response. Our recent findings showing that GC RAS signals can antagonize proliferative and differentiation processes[21] hinted at this potential. However, this phenomenon had passed unnoticed in our previous studies[9], likely because selective pressure had favored survival of the least pro-apoptotic lineages when generating stable cell lines. Arguably, a similar limitation may apply to zebrafish melanocytes, where we were unable to detect apoptosis. Consequently, we now demonstrate that a pro-apoptotic stimulus (TGF-β) activates the GC RAS pool, while proliferative (EGF) and differentiation (HRG) agonists do not, in line with previous reports showing that proliferative signaling entail RAS activation solely at the plasma-membrane[22,43,44].

In the same vein, we show that over-expression of RASGRP1, the GEF responsible for activating the RAS GC pool, can prevent ERK activation and induce apoptosis. Indeed, RASGRP1 has been shown to antagonize ERK activation previously[45] and has been associated with apoptotic processes in several cell lineages[46–49]. As RASGRP1 can activate RAS at multiple sublocalizations and can also intervene in proliferative processes[6,7], it is likely that the balance between survival and apoptogenic signals elicited by this GEF, probably at different sites, plays a major role in the life or death decision. However, since RASGRP1 is not expressed in the cell lines under scrutiny in this study it is likely that another, yet unidentified RAS GEF, is involved in conveying apoptotic stimuli to the RAS GC pool.

We also demonstrate that the accumulation of oncogenic RAS at the GC, both physiologically- and pharmacologically-evoked,

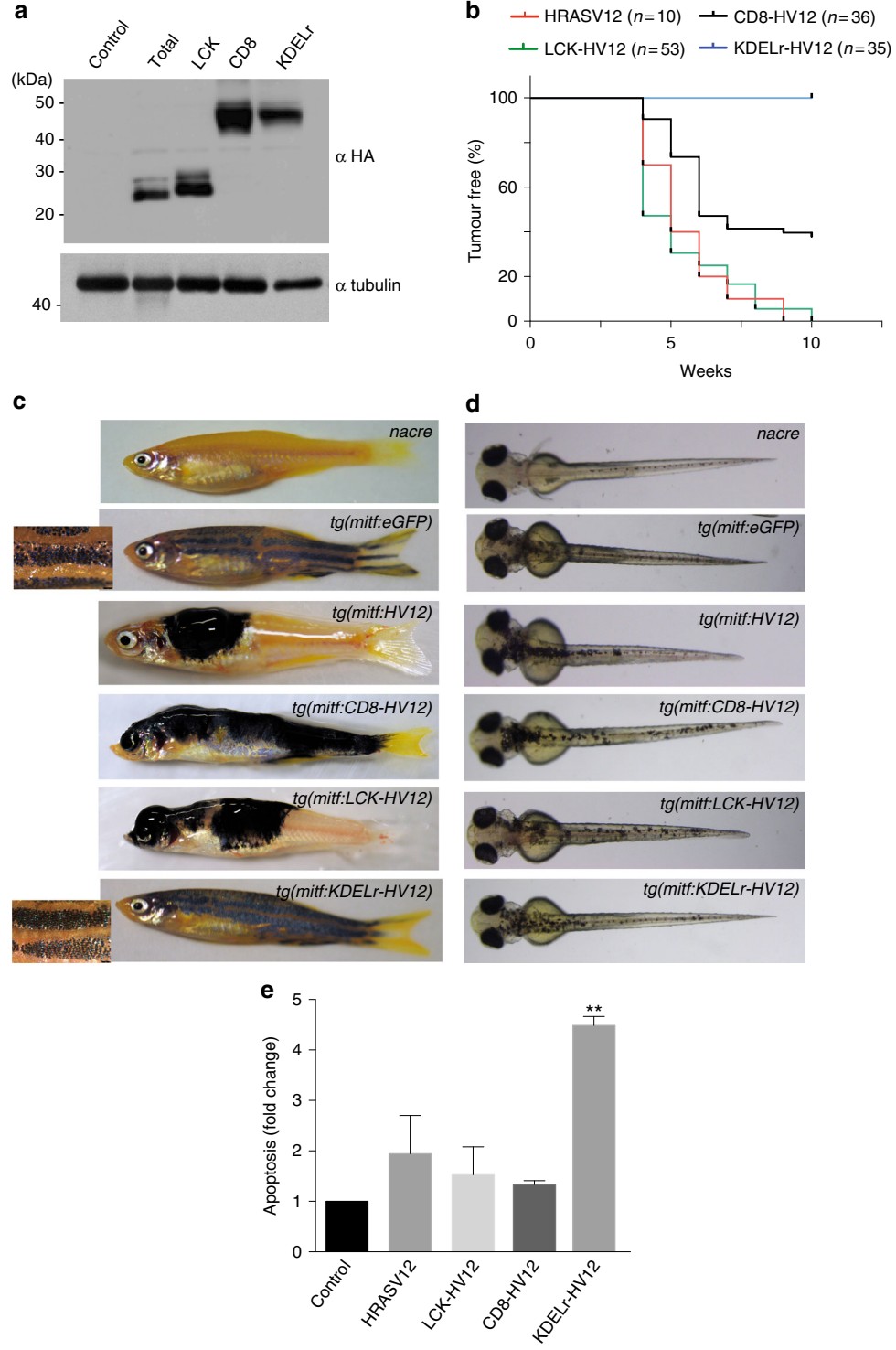

**Fig. 8** Oncogenic effects of activated RAS at the GC. **a** Expression of the indicated constructs transfected (1 μg) in CHL melanocytes. **b** Kaplan–Meier plot of the Incidence of melanoma by 10 weeks post fertilization in nacre zebrafish expressing the indicated HRASV12 site-specific transgenes using the miniCoopR system. **c** Representative images of adult zebrafish expressing the indicated HRASV12 site-specific transgenes at 8 weeks post fertilization compared to nacre. Insets are magnification of stripes. **d** Representative images of zebrafish embryos expressing the indicated HRASV12 site-specific transgenes at 3 days post fertilization compared to uninjected nacre. **e** Apoptosis induction by the indicated HRASV12 site-specific constructs in CHL melanoma cells. Results, relative to the values found in vector-transfected cells, show average ± SEM from three independent experiments **$p < 0.01$ by Student's $t$-test. See also Supplementary Fig. 7

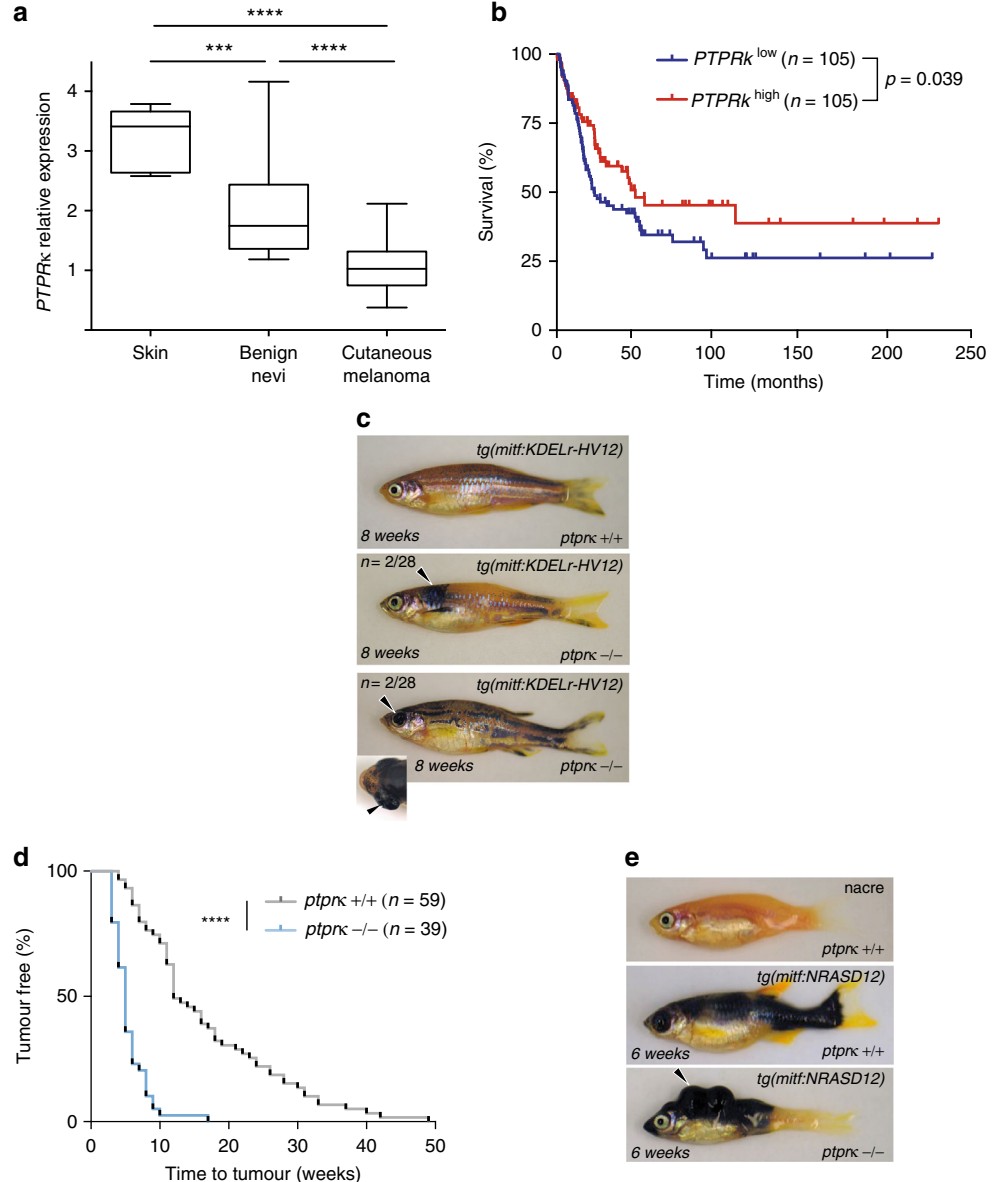

**Fig. 9** PTPRκ is a tumor suppressor in melanoma. **a** Analysis of *PTPR*κ expression in normal skin (*n* = 7), benign nevi (*n* = 18) and cutaneous melanoma samples (*n* = 45) generated using an available gene dataset, accessed through the Oncomine® platform. ****p* < 0.001 *****p* < 0.0001 by Student's *t*-test. **b** Survival curves for melanoma patients depending on PTPRκ expression levels (obtained from GEO GSE65904). **c** View of naevi (2/28; arrowhead, middle panel) and melanoma (2/28; arrowhead bottom panel, points to a protruding eye with a retroorbital tumor.) by 52 weeks post fertilization in *ptpr*κ nullizygous zebrafish expressing KDELr-HV12 as a transgene. Representative pictures were taken at 8 weeks post fertilization. **d** Kaplan–Meier plot of tumor development in *ptpr*κ wild-type nacre animals compared to *ptprk*−/− nacre animals, injected with NRAS-G12D. *****p* < 0.0001 by Mantel-Cox test. **e** Representative melanoma lesions (see arrowheads) in zebrafish with the indicated *ptpr*κ genotype expressing NRAS-G12D as a transgene. See also Supplementary Fig. 8

triggers apoptosis. In agreement, redistribution of oncogenic NRAS to endomembranes impairs haemopoietic cells growth[50] and oncolytic viruses cause accumulation of oncogenic HRAS in the GC to induce cell demise[51]. These results underscore the potential therapeutic value of redirecting oncogenic H/N RAS to the GC as an anti-neoplastic strategy.

We have also unveiled RALGDS as a mediator of GC RAS-induced apoptosis. Its overexpression does not induce apoptosis per se, but it does synergize with GC RAS to this end. RALGDS regulates survival in transformed cells[52], so probably RALGDS triggers pro- or anti-apoptotic responses depending on the sublocalization where it is activated. RAL GEFs activate JNK and p38 pro-apoptotic MAPKs via RAL GTPases[53,54]. Also, RAL GTPases can down modulate the survival factor NF-κB[55]. Thus, GC RAS-activated RALGDS could modulate these pathways to induce apoptosis. Of note, RASGRP1 downregulates of NF-κB to induce apoptosis in B cells[47].

We show that GC RAS induces apoptosis by down-regulating ERK activation via induction of *PTPR*κ. Indeed, we had previously detected this gene as one specifically switched-on by RAS signals emanating from the GC[11]. Moreover, we have unveiled that PTPRκ dephosphorylates CRAF at tyrosine residues essential for its activation[56,57]. However, since PTPRκ can inhibit other signaling intermediaries[36,58,59], we cannot exclude that the

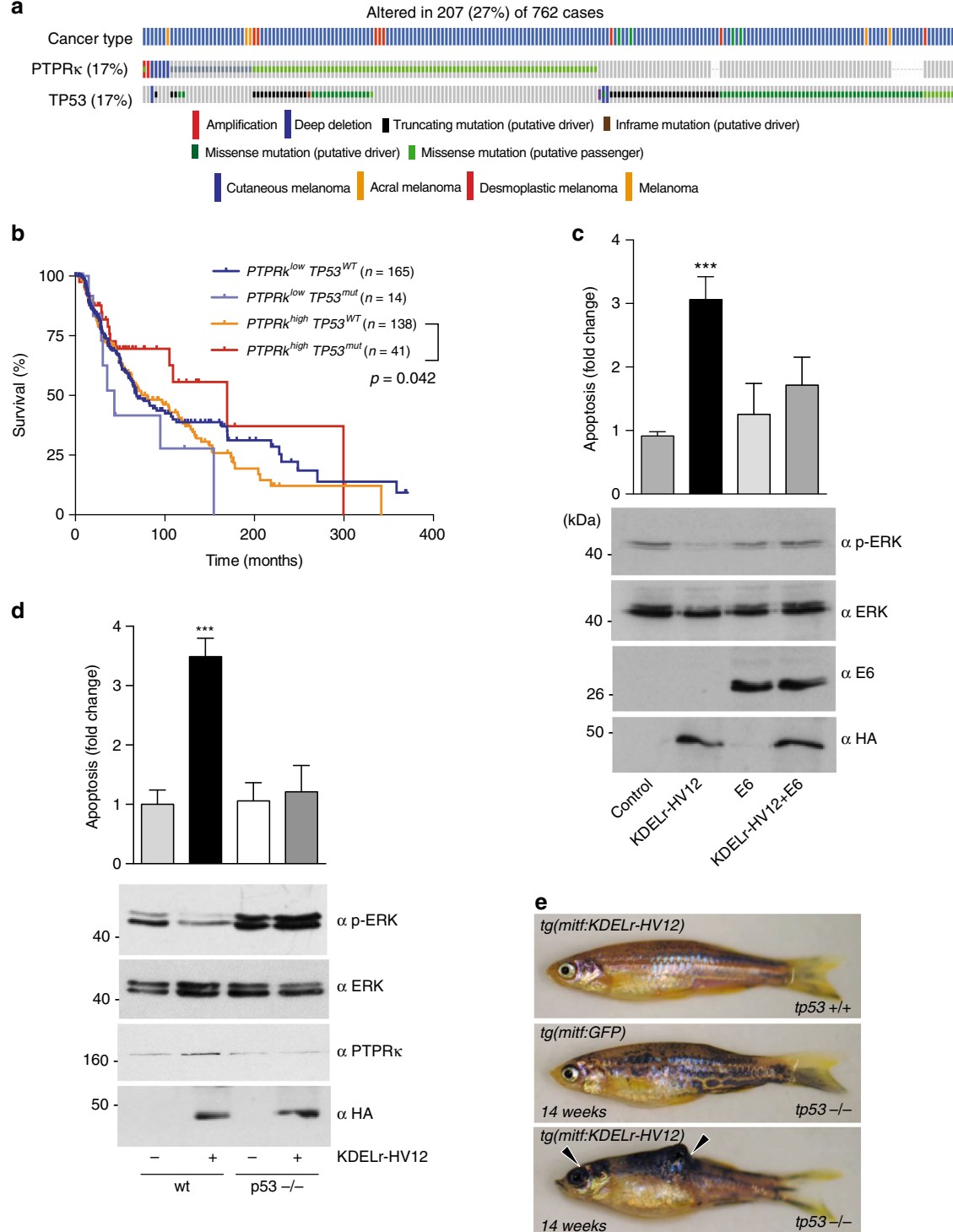

**Fig. 10** TP53 status determines GC RAS-induced melanomagenesis. **a** Analysis of *PTPRκ* and *TP53* mutational status generated using the TCGA dataset accessed through the cBioPortal for Cancer Genomics platform. **b** Survival curves for melanoma patients depending on *PTPRκ* expression levels and *tp53* mutational status (obtained from TCGA melanoma dataset). **c** Effects of TP53 inactivation on ERK phosphorylation and apoptosis induced by GC RAS signals, as revealed in MCF-7 cells transfected with the indicated constructs. Data show average ± SEM from three independent experiments. ***$p < 0.005$ by Student's *t*-test. **d** Effects of TP53 status on ERK phosphorylation and apoptosis induced by GC RAS signals, as revealed in MEFs, wild-type (wt) and tp53-null,transfected with KDELr-HV12 (1 μg) where shown (+). Data show average ± SEM from three independent experiments. ***$p < 0.005$ by Student's *t*-test. **e** tp53 inactivation is sufficient to promote melanomagenesis by GC RAS signals. Arrowheads show melanoma lesions in *tp53−/−* zebrafish expressing the KDEL-HV12 transgene after 14 weeks. See also Supplementary Fig. 9

repression of other processes also contributes to the apoptotic response. But, the ability to rescue PTPRκ-induced apoptosis by forced expression of MEKE or BRAFV600E certainly indicates that regulation of ERK signaling is of primary importance.

Using zebrafish, we demonstrate for the first time in an animal model that RAS oncogenic potential is strictly dependent on its sublocalization: oncogenic RAS induces melanoma when its signals emanate from the plasma-membrane, but not from the GC. Furthermore, disrupting ptprκ in zebrafish bolsters melanomagenesis by NRAS, consistent with its assigned tumor suppressor role[60,61]. Indeed, PTPRκ is downregulated in the majority of melanoma-derived cell lines[62] and in human patients cases as reported here, as is the case with other PTP family members[63]. In addition, PTPRκ loss can rescue to some extent GC RAS transforming potential. One possible explanation for this partial recovery is that ERK activation as elicited by GC RAS is of low intensity[9]. Thus, a more potent activation of the ERK pathway, as is the case for NRAS G12D, would be necessary. Though the possibility of additional oncogenic events, hitherto undisclosed, being required for full-blown malignancy cannot be discarded.

In this respect, we demonstrate that the absence of tp53 is sufficient to render GC RAS signals oncogenic, but coincident PTPRκ loss does not result in an enhanced carcinogenic effect. One possible explanation for this could come from recent findings showing that loss of TP53 leads to MEK/ERK activation by some unknown mechanism acting downstream of CRAF[64], thereby irrepressible by PTPRκ. Thus, concomitant loss of the phosphatase would not be expected to enhance ERK activation any further. A second explanation could be that ERK activation elicited by GC RAS being of low intensity[9], would not be bolstered by co-occurring tp53/ptprκ mutations much further than by each of them individually.

Overall, our findings reveal for the first time in an animal model that RAS oncogenic potential is strictly dependent on its sublocalization. As opposed to RAS prevailing role as an ERK activator, RAS signals emanating from the GC prevent ERK signaling via PTPRκ, a tumor suppressor that emerges as a novel regulator of this pathway, thereby inducing apoptosis. Thus, GC RAS anti-oncogenic effect could be exploited as a therapeutic venue in oncology.

## Methods

**Kinase assays**. Performed as described[65] using MBP or histone H2B as substrates. Two days after transfection, cells were cultured overnight in serum free medium. Kinase assays were performed essentially by an in vitro immunocomplex assay in anti-HA immunoprecipitates using myelin basic protein (MBP) (Sigma) for ERK2 kinase activity, AKT kinase activity was assayed using histone H2B as substrate.

**Immunoblotting and immunoprecipitations**. Performed as described[66], For immunoblotting, samples were fractionated by SDS-polyacrylamide gel electrophoresis and transferred onto nitrocellulose filters. Immunoprecipitations were performed in 20 mM HEPES, pH 8, 2 mM MgCl$_2$, 2 mM EGTA, 150 mM NaCl, and 2.5 mM CHAPS. Immunocomplexes were visualized by enhanced chemiluminescence detection (GE Healthcare, Little Chalfont, Buckinghamshire, United Kingdom) by using horseradish peroxidase-conjugated secondary antibodies (1:10,000 dilution) (Bio-Rad Laboratories, Hercules, CA). Mouse monoclonals anti: FLAG (1:10,000 dilution for WB) was from Sigma. Mouse monoclonals anti: HA (1:1000 dilution for WB), ERK2 (1:1000 dilution for WB) and phospho-ERK (1:1000 dilution for WB) from Santa Cruz. Rabbit polyclonals anti: ERK1/2 (1:1000 dilution for WB), MEK1 (1:1000 dilution for WB), PTPRκ (1:500 dilution for WB), RSK1 (1:2000 dilution for WB), phospho-MEK (1:500 dilution for WB), CRAF (1:500 dilution for WB), phospho-CRAF (1:500 dilution for WB) from Santa Cruz, HRAS (1:5000 dilution for WB) from Abcam. Mouse monoclonals anti: Caspase 3 (1:500 dilution for WB), MYC (1:1000 dilution for WB) from Cell Signaling; GM130 (1:500 dilution for IF) and TGN46 (1:400 dilution for IF) from BD Biosciences, Apt-1(1:500 dilution for WB) from Genway, RAS-GRP1 (1:500 dilution for WB) from Santa Cruz. Supplementary Fig. 10 shows the uncropped films corresponding to the panels displayed in the figures.

**Cell culture and transfection**. MCF-7, HeLa, CARC, T24, and other tumor cell lines cells (ATCC) were grown in Dulbecco's minimum essential medium (DMEM) supplemented with 10% fetal bovine serum. NIH3T3 (ATCC) in DMEM-10% calf serum. MEFs (Barbacid's Lab) in DMEM-10% fetal bovine serum. Cell lines were authenticated by the supplier. Cell lines were routinely tested for mycoplasma contamination by PCR. Where applicable, stable lines cells were generated by transfection with Lipofectamine (Invitrogen) following manufacturer's instructions and selected with 750 mg/ml G418 or 300 μg/ml zeocine (Invitrogen) where necessary. For biochemical analyses, subconfluent cells were transfected with Lipofectamine and Lipofectamine™ 2000 and 3000 (Invitrogen). For immunofluorescence studies, cells were transfected with FuGENE transfection reagent (Roche). Before stimulation, cells were starved for 18 h. TGF-β and UO126 were from Sigma, palmostatin B from Calbiochem- Merck-Millipore.

**Apoptosis**. The levels of apoptosis were quantified by evaluating either Caspase 3 activity by western blotting or using the Caspase-Glo 3/7 luminogenic assay (Promega) or Annexin V levels using the Guava/nexin assay (Millipore), in both cases following manufacturer's recommendations.

When performing the Guava Nexin (EMD Millipore Guava Technologies) assay, samples were gated with X and Y intercepts between 10 (10e1) and 30 (10e3) on a log-fold scale at apparent breaks in cell populations as illustrated in Supplementary Fig. 2. Once gated, cells within the lower left quadrant were not labeled with either marker, therefore are not undergoing detectable apoptosis, while cells within the lower right quadrant were positive for Annexin V-PE and negative for 7-AAD, marking them as early apoptotic cells. Those in the upper right quadrant were positive for Annexin V-PE and negative for 7-AAD, indicating late apoptosis. Very few cells were in the upper left quadrant and were not positive for the early apoptotic marker Annexin V-PE, so were not considered. Three experiments were independently plated, with four replicate wells of each cell type per experiment. Apoptosis results are represented as the fold change of the Caspase 3/7 activity or Annexin V levels, relative to the untransfected or untreated control cells.

**Time lapse immunofluorescence**. Cells were grown on polylysine-coated, glass-bottom dishes and transiently co-transfected with Cherry-HRASwt, or Cerulean-HRAS wt, GFP-Raf RBD E3-R3 (A/D)[22] and CellLight RFP-Golgi from Life Technologies. Cells were deprived of serum, placed into a microscope chamber and treated with agonists. Confocal images (512 × 512 pixels; 0.15 pixel size) were acquired at 37 °C in a TCS SP-5 confocal microscope (Leica) with a ×40, 1.25 NA oil objective, a 1 Airy pinhole and 200 Hz speed. Images were captured every 2 min. Cells were excited with 405 nm, 458 nm, and 543 nm laser lines. Images are presented after digital adjustment of brightness and contrast to maximize signal. Images were processed and analyzed using FIJI Image J. To quantify the degree of co-localization between fluorophores we used the Pearson correlation coefficient (PCC) using FIJI Image J.

**Confocal immunofluorescence**. Cultured cells were washed twice in PBS, fixed with ice-cold 3.7% formaldehyde in PBS for 10 min, and washed with PBS. They were rinsed in PBS-0.05% Tween 20 (Sigma-Aldrich), incubated for 2 h with the primary antibodies or GM130 antibody (BD Biosciences), washed, and incubated for 1 h with the appropriate secondary antibodies conjugated to FITC or Texas Red. Coverslips were mounted in VECTASHIELD-DAPI (Vector Laboratories, Burlingame, CA), and sealed. Confocal microscopy was performed with an LSM510 microscope (Carl Zeiss, Thornwood, NY), by using excitation wavelengths of 488 nm (for FITC) and 543 nm (for Texas Red). The images were then processed and analyzed using FIJI Image J.

**Subcellular fractionation**. A total of 100 mm$^2$ dishes of SKMEL2 cells were lysed in an isotonic buffer followed by sonication. Unbroken cells and debris were removed by a 10,000×g spin, and the extracts were fractionated by ultracentrifugation at 100,000×g to yield supernatant (S) and particulate (P) fractions. A total of 20 μg of each extract was purified by SDS-PAGE and HRAS was identified following immunoblotting.

**Cell cycle assays**. For cell cycle analyses, fixed cells were incubated in 5 μg/mL PI PBS1x for 20 min at room temperature and analyzed by MACSQuant VYB (Milteny Biotec) flow cytometer. Cell cycle was analyzed using MODFIT software.

**Focus forming assays**. Performed basically as described[67]. Briefly, subconfluent NIH3T3, cultured in DMEM-10% CS, were transfected with the indicated constructs using Lipofectamine reagent (Invitrogen), following manufacturer's instructions. After 10–15 days in culture plates were stained in 5% GIEMSA and foci were scored.

**Ras-GTP loading assays**. Ras-GTP loading assays were performed as described previously[67]. KDELr-H-Ras-GTP was affinity sequestered by using glutathione transferase -Raf-RBD. Immunoblots were performed with anti-HA antibody and quantified by densitometry using Image J. Activation levels were related to total protein levels as determined by anti-HA immunoblotting in the corresponding total lysates.

**Senescence assay**. Expression of pH-dependent senescence-associated β-galactosidase (SA-β-gal) activity was analyzed using the SA-β-gal staining kit (Cell Signaling Technology, Boston, MA). Quantification of the SA-beta-gal positive cells were done using light microscopy (×63 magnification). At least 200 cells were counted per condition. Results are represented as the fold change of the SA-β-gal positive cells, as compared to the untransfected or untreated control.

**Statistical analyses**. Throughout, graphed data are expressed as mean ± standard error of the mean (SEM). Each dataset's descriptive statistics and distribution was analyzed using the explore function in SPSS software. For experiments involving cultured cells, unless otherwise stated values are expressed as means ± SEM of three independent experiments; $P$-values were calculated with the two-tailed Student's $t$-test, 95% significance. To test for the homogeneity of variances we run a previous $F$-test of equality of variances. Significance was assessed using parametric or non-parametric tests as appropriate. Statistical significance was set to $p < 0.05$.

**Zebrafish care and maintenance**. Zebrafish were housed at the Biological Services Unit (BSU), The University of Manchester, and maintained at 28.5 °C under a 14 h light/10 h dark cycle. Embryos were collected and raised in egg water (instant ocean salt 60 g/ml and methylene blue 0.5 mg/ml) at 28.5 °C up to 5 days post fertilization and thereafter on a recirculating system fed with live and powdered brine shrimp. Zebrafish husbandry and experimental studies conducted at The University of Manchester were performed in compliance with the Animals (Scientific Procedures) Act 1986 under a Home Office approved project license.

**Melanoma induction in zebrafish**. Middle entry (ME) vectors containing HRASV12, KDELr HRAS-V12, LCK-HRASV12, CD8-HRASV12, NRAS-D12, and BRAF-E600, all of human origin, were created using primers with *att*B sites (see Supplementary Table 2 for primer sequences) to amplify appropriate cDNA templates and then recombined into pDONR221 by performing a recombination reaction catalyzed by BP clonase II Plus enzyme mix (Life Technologies) according to the manufacturer's instructions. Individual transgene vectors were then created by a Multisite Gateway recombination reaction catalyzed by LR clonase II Plus enzyme mix (Life Technologies) according to the manufacturer's instructions, combining the ME clone of interest with p5′E-*mitfa* promoter, p3′E-polyA entry clones and mini-CoopR destination vector[38] all kindly provided by Dr Craig Ceol (UMass Medical School). All constructs were verified by sequencing and restriction digestion (plasmid sequences available on request). A total of 40 picograms (pg) of each transgene together with 40 pg of Tol2 transposase mRNA [synthesized from a NotI-linearized pCS2-TP plasmid, a kind gift from Dr. Koichi Kawakami (National Institute of Genetics), using the SP6 mMessage mMachine kit (Ambion) then purified using RNeasy Mini Kit (Qiagen)] were microinjected into the yolk of 1-cell stage zebrafish embryos derived from *nacre* (*mitfa*$^{w2}$) mutant or *nacre*; *tp53*$^{M214K}$ mutant animals as indicated in each figure. Equal level of pigment rescue was determined 3 days post fertilization, and the selected animals scored weekly for the presence of visible tumors. Animals were randomly assigned to different experimental groups but no formal method of randomization was used. Blinding was not used when scoring phenotypes. Power analysis (using Graphpad statmate) of pilot experiments informed minimum samples size. The Kaplan–Meier method was used to calculate the probability of tumor formation and significance established using a Mantel-Cox test (Prism 7.3; GraphPad Software) with the significance set at $p \leq 0.05$.

**Real-time quantitative PCR**. RNA was isolated from homogenized embryos using TRIzol (Life technologies). Reverse transcription was carried out using the ProtoScript II first-strand cDNA synthesis kit (New England Biolabs) with oligo(dT) primers. Real-time quantitative PCR (qPCR) was performed using SYBR Green JumpStart *Taq* ReadyMix (Sigma-Aldrich) and the MX300P system (Stratagene) with a 60 °C annealing temperature. *ptprk* expression was normalized to elongation factor 1 alpha.

**Cloning zebrafish *ptprk***. The cDNA was PCR amplified from 1-cell stage *nacre* embryo cDNA using the primers described in Supplementary Table 2. The PCR product was directionally cloned into pCS2+MfeI/SpeI sites to generate zptprk-pCS2+. The construct was verified by nucleotide sequencing using the primers in Supplementary Table 2. The nucleotide sequence is available from GeneBank: MG189366.

**Genome editing**. Capped nls-zCas9-nls mRNA was synthesized using a mMES-SAGE mMACHINE SP6 kit (Life Technologies) from a pCS2 construct (plasmid #47929; addgene) and purified using a RNeasy mini kit (Qiagen). Zebrafish *ptprk* targeting guide was designed using the Harvard chopchop program (https://chopchop.rc.fas.harvard.edu). Guide RNA (gRNA) incorporating this target sequence was generated from a PCR amplification product (see Supplementary Table 2 for primer sequences) including the remaining sequence of *S. pyogenes* chimeric single guide RNA through in vitro transcription using a T7 RNA polymerase MEGA short script T7 kit (Life Technologies). The gRNA was then precipitated in a 1/10 volume of 3M Sodium Acetate and two volumes of 100% ethanol by chilling the reaction at −20 °C for 15 min, then spinning in a microcentrifuge (sigma) at 13K for 15 min, and

finally the RNA pellet was resuspended in 15 µl of RNase free water. Cas9 mRNA (250 pg) and gRNA (30 pg) were injected into the yolk of 1-cell stage embryos. Working guides were identified by PCR amplifying the target region and running the PCR product on a 3% agarose gel to identify INDEL events that produce visible shifts or smearing of the amplification product.

**Genotyping**. Embryos or fin-clips were placed in PCR tubes with 50 µl of 50 mM NaOH and denatured for 20 min at 95 °C. A volume of 20 µl of Tris-HCl pH 8 was added to each tube and 1 µl of the genomic DNA used for PCR amplification (primer sequences can be found in Supplementary Table 2).

**Polymerase chain reaction conditions**. Polymerase chain reaction (PCR) was performed in a 25–50 µl reaction mix containing DNA template (0.1–100 ng DNA), sense and antisense primer 0.8µm each, 0.25 mM dNTPs (Bioline), 1X HF buffer (New England Biolabs), 1U Phusion Taq polymerase (New England Biolabs), 0.5 mM MgCl (New England Biolabs) and 1.5 µl DMSO (New England Biolabs) per 50 µl reaction. PCR was performed in a Techne TC-PLUS or Alpha Thermal Cycler PCR$^{MAX}$ machine with an initial denaturing step at 98 °C for 3 min followed by 35 cycles of denaturing at 98 °C for 10 s, annealing at 60 °C for 30 s, and amplification at 72 °C for 1 min/1 kb. A final 10 min cycle at 72 °C was routinely performed to allow the complete extension phase to occur.

**Imaging**. Zebrafish adults were anesthetized using MS-222 (Sigma-Aldrich) and imaged using a Nikon digital D3000 camera with a AF-S micro nikkor 105 mm 1:2.8 G ED lens

**Gene-expression analysis using the oncomine$^{TM}$ platform**. The Talantov melanoma dataset[39] containing 70 samples: 7 skin, 18 benign melanocytic skin nevi and 45 cutaneous melanoma samples was exported from Oncomine and analyzed in GraphPad Prism 7.3 (GraphPad Software). Data were analyzed using a Student's $t$-test and the significance threshold was set at $p \leq 0.05$.

**Bioinformatics of human melanoma sequencing data**. R version 3.3.1 was used for the statistical analyses along with Perl for text processing. Survival analyses were performed through Kaplan–Meier estimates of the overall survival according to the expression level of *PTPRκ*. The PTPRκ$^{low}$ and PTPRκ$^{high}$ groups were established using the median of the distribution of expression values as a split point. The Mantel-Cox test was applied to statistically validate the differences between the survival distributions of such groups. The accession code for the Gene Expression Omnibus (GEO) melanoma dataset used in this study is GSE65904.

## Data availability

The datasets generated during and/or analysed during the current study are available from the corresponding author on reasonable request

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

## Acknowledgements

We are grateful to Drs: Ignacio Rubio, Yardena Samuels, Mariano Barbacid and Javier León for providing reagents; and Alicia Noriega, Sandra Zunzunegui y Victor Campa for technical support. Crespo laboratory is supported by grant SAF-2015 63638R (MINECO/FEDER, UE); by Red Temática de Investigación Cooperativa sobre el Cáncer (RTICC). RD/12/0036/0033 and by Asociación Española Contra el Cáncer (AECC), grant GCB141423113. Work in the Hurlstone laboratory was funded by a grant from the European Research Council (ERC-2011-StG-282059 PROMINENT). B.C. is supported by a Retos Jóvenes Investigadores grant SAF2015-73364-JIN (AEI/FEDER, UE) and a grant from Fundación Francisco Cobos. X.R.B. is supported by grants from the Castilla-León Government (BIO/SA01/15, CSI049U16), MINECO (SAF2015-64556-R, RD12/0036/0002), Worldwide Cancer Research (14-1248), Ramón Areces Foundation, and AECC (GC16173472GARC). Spanish funding to P.C., B.C., and X.R.B. is partially supported by the European Regional Development Fund.

## Author contributions

B.C. and A.P.B. performed most of the experiments with the exception of: Fig. 4f, g and Fig. 3c, d (I.J.); Fig. 3a, b (I.A.); Fig. 6c (P.C.-B.); Fig. 3e (V.C.); J.B. and I.B.-R. contributed in the experiments with zebrafish. D.J.D. provided reagents. L.F.L.-M. and X.R.B. performed bioinformatics analysis on human tumor samples and contributed to manuscript writing. B.C. and A.P.B. also prepared the figures and performed the statistical analyses. P.C. and A.H. conceived the study, directed it, analyzed and interpreted data and wrote and corrected the manuscript.

## Additional information

**Competing interests:** The authors declare no competing interests.

