## [Peer Review File · Nature Communications]

Reviewer #1 (Remarks to the Author):

This is an interesting study which links the oncogenic activity of H/NRAS to its localization at the GV vs. plasma membrane. Their data suggests a model in which the GC tethered form of RAS can actually inhibit oncogenesis, in part by inhibition of ERK and subsequent apoptosis. Overall, the data are convincing and open up several new questions about RAS biology, some of which might even have therapeutic implications. The areas that would improve the clarity of their findings:

1) In Figure 1, the authors use MCF7 cells to show that GC RAS can be induced by apoptotic stimuli such as TGF β . In Figure 1b and 1d, the authors state that the activity is limited to the periphery with EGF but the GC with TGF β . But the figures need better annotation of cell structures to make sense of this. A true plasma membrane marker, along with a nuclear marker, would all be helpful to fully convince that this localization is to the locations they state – from these figures it is very hard to tell.

2) The authors use palmostatin to prevent RAS cycling, which is a good demonstration of the role of acylation. However, they do not show that at the doses they are using that the compound truly prevents cycling in the manner they describe. They should show this data to indicate the drug works as they think. A complementary approach would be to use RNAi of APT1 to show you affect shuttling and the subsequent apoptosis effects.

3) The temperature shift experiments are somewhat confusing and hard to interpret. The authors seem to be arguing that it is accumulation of GC RAS, rather than depletion of PM RAS, that induces apoptosis. But while it is true that shifting to the lower temp will prevent transport, a multitude of things are also changed at such a dramatic temperature shift. It is hard to conclude that “accumulation of mutant RAS at GC was indeed responsible for triggering apoptosis” based on this data. If the goal of this experiment was to show that it was not depletion of the plasma membrane pool of RAS, then the authors will need to genetically knock out RAS and rescue back with either plasma membrane vs. GC targeted RAS and measure apoptosis. It is possible that it is both effects that contribute to apoptosis, but this needs to be more rigorously shown that with the temp shift experiments.

4) The data in Figure 3 links loss of ERK phosphorylation to the GC form of RAS, which is an interesting finding. They then go on to suggest that it is the interference with pERK that is responsible for the apoptotic effect. However, this is not clear to me that this is what loss of pERK is doing. Wouldn't the loss of pERK elicit a potent effect on either cell cycle and/or a senescence response in the cells? In other words, is the apoptosis you are seeing a byproduct of prolonged cell cycle arrest and/or senescence, and not a direct effect on apoptosis at all? This could be shown using

standard assays for these effects. Moreover, maneuvers to manipulate cell cycle (i.e. CDK4/6i) could demonstrate whether the effect of GC RAS is ultimately mediated via this effect.

5) Is it known whether the effect of PTPRK on CRAF phosphorylation is direct or indirect? The co-IP data is suggestive but not conclusive – is there any data the authors have which would suggest a more direct effect?

6) The zebrafish data in Figure 6 is an interesting in vivo correlate of their findings and strengthens their conclusions. However, one confusing finding is that while the KDEL-RAS fails to induce tumors, it is also expressed at much lower levels, and it is also not clear to me what tissue was used for this Western blot. While we recognize that these are very time consuming experiments, isn't another explanation of the findings in 6b that they never expressed enough of the protein to get over the threshold needed for tumorigenesis? Another unexplained finding here is that in the rest of the paper, the authors argue that GC RAS induces apoptosis, yet in looking at the KDEL fish in Figure 6c, it looks they have a full complement of melanocytes, arguing against an apoptotic event that prevents their development. These issues need to be addressed to fully make sense of the data in figure 6.

7) In Figure 7, it is stated that the KDEL NRAS "occasionally" gives rise to nevi and/or a possible melanoma, but there are no numbers here to quantify, which would be helpful especially since in Figure 6 the KDEL looks like it never gives rise to tumors. But more importantly, what isn't clear to me in this experiment is the p53 status of the animals. In all of the other fish studies, it looks like p53 is lost, which probably boost the tumorigenic capacity of the system. But here it looks like p53 might be WT, so this raises the question more broadly then of the role of p53 in all of the findings of the paper. Do the authors have data to indicate to what extent the effects of GC RAS are p53 dependent or not? This question should be addressed in terms of both their in vitro cell line data as well as their in vivo zebrafish data.

Reviewer #2 (Remarks to the Author):

The partition of NRAS and HRAS between the plasma membrane (PM) and Golgi apparatus is regulated by palmitoylation/depalmitoylation and it is now well-established that the steady-state distribution of these RAS proteins includes both subcellular compartments. This has led RAS

biologists to question whether RAS signaling can be initiated from both of these compartments and if there are functional differences that constitute compartmentalized signaling. Although multiple groups have confirmed compartmentalized signaling, some of the details remain unresolved and some reports are contradictory. For example, whereas forced overexpression of constitutively active HRAS tethered to the Golgi apparatus with the KDELr was initially reported to stimulate the MAPK pathway (Ref. 14), transform cells (Ref. 14) and induce differentiation of PC12 cells (Ref. 6), Crespo and colleagues reported contradictory results (Ref. 9). Perhaps most compelling were reports of RAS signaling in lymphocytes where MAPK signaling could be initiated from the Golgi (Ref. 16) and, most important, signaling from the PM stimulated negative selection (cell death) while signaling from the Golgi drove positive selection (proliferation) (PMID: 17086201). Casar et al. now add to the complexity with a study that purports to show that RAS activation on the Golgi blocks tumorigenesis by inducing apoptosis via PTPRK-mediated inhibition of ERK activation. Were the data compelling this would be an interesting and potentially important report. Unfortunately, the data do not support all of the conclusions.

The story begins with the observation that KDELr-HRASV12 can be loaded with GTP by overexpression of RASGRP1 or stimulation with TGF- β but not EGF or HRG. This is supported by live cell imaging of RAS activation using a fluorescent probe. The authors interpret the data shown in Fig. 1b and d as showing activation by EGF on the PM and by TGF- β on the Golgi but the data do not support this conclusion. First, contrary to most of the literature cited, mCherry-HRAS is not seen on the Golgi making it unlikely that activation on that compartment can be assessed. Second, the cell scored for TGF- β induced activation of HRAS on the Golgi has an atypical and uninterpretable distribution of mCherry-HRAS and a distribution of RAF-RBD-GFP in the nucleus and cytosol without any Golgi like structure decorated in the green channel. The area of overlap that is interpreted as Golgi is simply the area of overlap of cytosolic RAF-RBD-GFP with the odd distribution of mCherry-HRAS, which reads with a strip of yellow in the oversaturated micrograph. For TGF- β induced signaling on the Golgi to be established the authors must show several additional clear examples, preferably in more than one cell type, with statistics of number of cells examined, and validate activation on the Golgi with a marker protein such as those employed in Fig. S2c.

The central conclusion hinges on the idea that the signaling capabilities of KDELr-HRASV12 differs from that of HRASV12 only because of the former's constitutive localization on the Golgi apparatus. These differences are reported as stimulation of apoptosis and inhibition of ERK signaling. The data for the latter are compelling and very interesting, however the mechanism is uncertain. An alternate Golgi tether, SCG10 did not impart the same properties on HRASV12 as did KDELr, a finding the authors interpret as indicating that inhibition of ERK ensues exclusively from the cis-Golgi. However, this calls into question the idea that the differences in signaling are exclusively related to subcellular localization and require further analysis with additional tethers and linkers of different lengths (RAS to Golgi membrane). Other explanations are possible. Natively targeted HRAS17N is a potent dominant negative signaling element in the RAS/MAPK pathway because it is capable of sequestering GEFs. Might KDELr-HRASV12 sequester some element of the signaling pathway that

does not represent physiologic signaling? Indeed the data presented suggest that KDELR-HRASV12 binds preferentially to RALGDS.

Most of the data presented are results of overexpression of signaling molecules. The only data to suggest that a similar phenomenon operates for endogenous RAS are shown in Fig. 3b where overexpression of an artificially Golgi targeted GEF downregulates ERK signaling in response to EGF or overexpression of HRAS12V. The physiologic correlate would be RASGRP1 signaling since it is the RAS GEF known to function at the Golgi. But RASGRP1 potently stimulates ERK signaling (see for example PMID: 17283063). How do the authors reconcile this contradiction? Have the authors considered examining RASGRP1 deficient cells? Is GTP loading of HA-KDELR-HRAS in response to TGF- β (Fig. 1c) dependent on RASGRP1? Does the effect of PalmB on selected tumor cell lines (Fig. 2d,f) depend on RASGRP1?

The SCG10-HRAS12V control shown in Supplemental Fig. 2 is compelling and underutilized. It would be informative to employ this construct in virtually all of the experiments, particularly the zebrafish studies shown in Fig. 6 and 7.

How do the authors explain the paradoxical behavior of HeLa cells when KDELR-HRAS12V is overexpressed? Does this mean that the wiring of KDELR-HRAS12V as a negative regulator of ERK signaling and a positive regulator of apoptosis is cell type specific? How so?

In Fig. 6c, why are the fish expressing KDELR-HRAS12V from a *mitfa* promoter more pigmented relative to the eGFP control, albeit in the absence of tumor? Does this not indicate proliferation rather than apoptosis of melanocytes?

Other specific comments:

It is not clear how apoptosis was scored. Results are reported as “fold change.” What does this mean? Was it the percentage of cells above some fluorescent annexin V gate in cytofluorometric analysis? How was that gate established?

We are asked to make a leap of faith that Palm B or exposure to 21°C mislocalizes RAS to the Golgi in a number of tumor cell lines (Fig. 2d,e). This must be shown experimentally. Also, it is paradoxical that Palm B, which putatively inhibits depalmitoylation, promotes accumulation of HRAS on the

Golgi. Also, Palm B has many off-target effects (PMID: 26701913). In contrast, 2-bromopalmitate (2-BP), which inhibits protein palmitoylation, allows NRAS and HRAS to accumulate on the Golgi for reasons that are easy to understand. What are the effects of 2-BP in this system?

The blot of melanoma cells in Fig. 2e suggests that an HRAS-specific antibody was utilized. This is not likely the case since melanoma cells express relatively little HRAS and HRAS, in contrast to NRAS, has not been reported in cell supernatants (see for example PMID 27502489). Also the arrows referred to in the legend are not shown.

None of the immunoblots shown indicate MW. Whereas this is okay for routine demonstration of proteins like MEK, ERK and Tubulin, where fusion versus endogenous proteins come into play (e.g. Fig. 1, 3c,d, 6a) MW markers would be helpful.

When results are shown as immunoblots the authors should indicate of how many independent experiments are the blots shown representative.

For many of the experiments shown (e.g. Fig. 2a, b, c, 3e, 5b) the relative expression level of the expressed proteins is critical to the interpretation of the results and must be shown.

In Fig. 7a is there a way to stratify the melanoma data by genotype? It would be informative to parse the expected 20% that are NRAS mutant from the majority that are BRAF mutant.

REVIEWERS REBUTTAL

1) In Figure 1, the authors use MCF7 cells to show that GC RAS can be induced by apoptotic stimuli such as TGFB. In Figure 1b and 1d, the authors state that the activity is limited to the periphery with EGF but the GC with TGFB. But the figures need better annotation of cell structures to make sense of this. A true plasma membrane marker, along with a nuclear marker, would all be helpful to fully convince that this localization is to the locations they state – from these figures it is very hard to tell.

Agree. In Fig 1b we have brightened the cherry RAS signal so endomembranes are visible and we have incorporated a nuclear marker (DAPI). In the inset absence of RAS activation at endomembranes and RAS activation at the plasma-membrane can be clearly differentiated, so we believe that in this case a Golgi marker is unnecessary. Following the reviewers advise, in Fig 2b we have incorporated a Golgi marker (RFP Golgi) to distinguish this organelle from plasma-membrane. We believe that RAS activation at this organelle is now clear. In addition, in Supp Fig 1 two more instances have been included, showing a profuse RAS activation at the Golgi concomitant with TGFB-induced apoptosis. Noticeably, RAS activation at the GC is more profuse in “ugly” cells, where the apoptotic process is more advanced.

2) The authors use palmostatin to prevent RAS cycling, which is a good demonstration of the role of acylation. However, they do not show that at the doses they are using that the compound truly prevents cycling in the manner they describe. They should show this data to indicate the drug works as they think.

Agree. In Supp Fig 3 we now incorporate immunofluorescences for A375 and T24 cells showing that palmostatin B and 21°C treatments result in a profuse RAS accumulation at the Golgi complex.

A complementary approach would be to use RNAi of APT1 to show you affect shuttling and the subsequent apoptosis effects.

Agree. This is a nice experiment. We have incorporated it in Fig 3C, showing that, indeed, APT1 depletion causes apoptosis.

3) The temperature shift experiments are somewhat confusing and hard to interpret. The authors seem to be arguing that it is accumulation of GC RAS, rather than depletion of PM RAS, that induces apoptosis. But while it is true that shifting to the lower temp will prevent transport, a multitude of things are also changed at such a dramatic temperature shift. It is hard to conclude that “accumulation of mutant RAS at GC was indeed responsible for triggering apoptosis” based on this data. If the goal of this experiment was to show that it was not depletion of the plasma membrane pool of RAS, then the authors will need to genetically knock out RAS and rescue back with either plasma membrane vs. GC targeted RAS and measure apoptosis. It is possible that it is both effects that contribute to apoptosis, but this needs to be more rigorously shown that with the temp shift experiments.

Agree. Previous data from Barbacid’s lab shows that genetic ablation of the three RAS isoforms, therefore complete depletion of the plasma-membrane RAS pool, is not sufficient to elicit apoptosis (PMID: 20150892). In Supp Fig 4B we show that these “Ras-less” cells undergo apoptosis when transfected with a GC-targeted but not with a PM-targeted RASV12, demonstrating that it is the accumulation of RAS at the Golgi rather than its absence from the PM what triggers apoptosis.

4) The data in Figure 3 links loss of ERK phosphorylation to the GC form of RAS, which is an

interesting finding. They then go on to suggest that it is the interference with pERK that is responsible for the apoptotic effect. However, this is not clear to me that this is what loss of pERK is doing. Wouldn't the loss of pERK elicit a potent effect on either cell cycle and/or a senescence response in the cells? In other words, is the apoptosis you are seeing a byproduct of prolonged cell cycle arrest and/or senescence, and not a direct effect on apoptosis at all? This could be shown using standard assays for these effects.

Agree. In Supp Fig 4C and D we show that the presence of activated RAS at the Golgi does not induce senescence neither alterations in the cells cycle. Thus, in these cells, interference with ERK activation results solely in apoptosis. ERK inhibition, for example by the MEK inhibitor U0126, can result in apoptosis in the absence of cell cycle arrest (For example PMID: 21163924).

5) Is it known whether the effect of PTPRK on CRAF phosphorylation is direct or indirect? The co-IP data is suggestive but not conclusive – is there any data the authos have which would suggest a more direct effect?.

Unfortunately, generating that data poses an extreme technical challenge. Demonstrating such direct effect requires an in vitro dephosphorylation assay that would take: 1) purifying full-length CRAF: impossible in bacteria, extremely low yields in baculovirus. 2) Phosphorylating it in vitro: who phosphorylates CRAF in the tyrosine/s dephosphorylated by PTPRK: SRC?, TAK?, other?. 3) Purifying full-length PTPRK: again extremely difficult as a consequence of the hydrophobicity of its transmembrane domain, and using only the cytoplasmic, phosphatase domain would deprive the protein of its "personality", making the result foreseeable yet unreal. Even though, we cannot provide a definitive answer, the results in Fig 6F showing that increasing concentrations of PTPRK induce CRAF tyrosine dephosphorylation and those on Fig 6G showing a direct interaction between both proteins, strongly suggest a direct effect. Though PTPK activating another phosphatase, though unlikely, cannot be absolutely discarded.

6) The zebrafish data in Figures 6 is an interesting in vivo correlate of their findings and strengthens their conclusions. However, one confusing finding is that while the KDEL-RAS fails to induce tumors, it is also expressed at much lower levels, and it is also not clear to me what tissue was used for this Western blot. While we recognize that these are very time consuming experiments, isn't another explanation of the findings in 6b that they never expressed enough of the protein to get over the threshold needed for tumorigenesis?

Agree. Former Figure 6 was rather confusing as the KDEL-RAS signal was lower than those of the other RAS constructs, but so were the corresponding tubulin levels. We now show the expression of the different constructs in CHL melanocytes, in which it can be noticed that KDEL-RAS is expressed to levels comparable to those of total and LCK-RAS, which induce massive melanomas, speaking against low KDEL-RAS levels as an explanation for its non-tumorigenicity. In addition, we now show that KDEL-RAS can induce tumors in the absence of PTPRK (Fig 9) or of p53 (Fig 10) demonstrating that its expression levels are sufficient for oncogenic transformation given the appropriate genetic setting.

Another unexplained finding here is that in the rest of the paper, the authors argue that GC RAS induces apoptosis, yet in looking at the KDEL fish in Figure 6c, it looks they have a full complement of melanocytes, arguing against an apoptotic event that prevents their

development. These issues need to be addressed to fully make sense of the data in figure 6.

Agree. In Figure 8D we now show that at early embryological stages fish display similar melanocyte numbers irrespective of the site where transgenic HRASV12 was targeted. In addition, in adult fish the stripes appear normal (insets Fig 8C). This could suggest that, unlike human melanocytes where it induces a potent apoptosis (Fig 8E), in zebrafish melanocytes KDELRASV12 does not induce apoptosis. However, we cannot discard that from very early embryonic stages selective pressure will favour survival and take-over by the least pro-apoptotic melanocytic lineages, in such a way that detecting apoptosis or a drop in the numbers of melanocytes in adults would be impossible. What seems very clear, is that GC RAS is non-tumorigenic, under normal conditions, irrespective of the model.

7) In Figure 7, it is stated that the KDEL RAS “occasionally” gives rise to nevi and/or a possible melanoma, but there are no numbers here to quantify, which would be helpful especially since in Figure 6 the KDEL looks like it never gives rise to tumors.

Agree. 2/28 fish, in both cases, displayed lesions.

But more importantly, what isn't clear to me in this experiment is the p53 status of the animals. In all of the other fish studies, it looks like p53 is lost, which probably boost the tumorigenic capacity of the system. But here it looks like p53 might be WT, so this raises the question more broadly then of the role of p53 in all of the findings of the paper. Do the authors have data to indicate to what extent the effects of GC RAS are p53 dependent or not? This question should be addressed in terms of both their in vitro cell line data as well as their in vivo zebrafish data.

Agree. We now provide data showing that, indeed, p53 status is critical for GC-RAS induced effects. In MCF-7 cells, we show that inhibiting p53 by the E6 protein precludes GC-RAS-induced ERK inactivation and apoptosis (Fig 10C). Likewise, GC-RAS does not inhibit ERK activation nor induce PTPK expression and apoptosis in p53 -/- fibroblasts (Fig 10D). And we show that in p53-null fish GC-RAS induces melanomas (Fig 10E). We are very grateful for this suggestion as it has introduced a very interesting twist in our findings.

Reviewer #2 (Remarks to the Author):

(.....) The authors interpret the data shown in Fig. 1b and d as showing activation by EGF on the PM and by TGF- β on the Golgi but the data do not support this conclusion. First, contrary to most of the literature cited, mCherry-HRAS is not seen on the Golgi making it unlikely that activation on that compartment can be assessed.

Agree. We apologize for not providing sufficiently clear images. In Fig 1B we have now brightened the cherry RAS signal so endomembranes are visible. In the inset, absence of RAS activation at endomembranes and RAS activation at the plasma-membrane can now be clearly differentiated.

Second, the cell scored for TGF- β induced activation of HRAS on the Golgi has an atypical and uninterpretable distribution of mCherry-HRAS and a distribution of RAF-RBD-GFP in the nucleus and cytosol without any Golgi like structure decorated in the green channel. The area of overlap that is interpreted as Golgi is simply the area of overlap of cytosolic RAF-RBD-GFP with the odd distribution of mCherry-HRAS, which reads with a strip of yellow in the oversaturated micrograph. For TGF- β induced signaling on the Golgi to be established the authors must show several additional clear examples, preferably in more than one cell type, with statistics of number of cells examined, and validate activation on the Golgi with a marker protein such as those employed in Fig. S2c.

Agree. Following the reviewers advise, in Fig 2B we have incorporated a Golgi marker (RFP Golgi) to distinguish this organelle from plasma-membrane. We believe that RAS activation at this organelle is now clear. In addition, in Supp Fig 1 two more instances have been included, showing a clear and profuse RAS activation at the Golgi, concomitant with TGF β -induced apoptosis. Of note: Noticeably, RAS activation at the GC is more profuse in “ugly” cells, where the apoptotic process is more advanced. Statistics of the fields observed and cell numbers have also been incorporated.

The central conclusion hinges on the idea that the signaling capabilities of KDELR-HRAS V12 differs from that of HRAS V12 only because of the former’s constitutive localization on the Golgi apparatus. These differences are reported as stimulation of apoptosis and inhibition of ERK signaling. The data for the latter are compelling and very interesting, however the mechanism is uncertain. An alternate Golgi tether, SCG10 did not impart the same properties on HRASV12 as did KDELR, a finding the authors interpret as indicating that inhibition of ERK ensues exclusively from the cis-Golgi. However, this calls into question the idea that the differences in signaling are exclusively related to subcellular localization and require further analysis with additional tethers and linkers of different lengths (RAS to Golgi membrane).....

Agree. We thank the reviewer for this suggestion as it was a necessary control. We have now performed additional experiments with another cis-Golgi tether (Supplementary Figure 6B,C): the avian infectious bronchitis virus E1 protein (PMID 10918587). When targeted to the GC by this means activated HRAS proved as efficient as KDELR-tethered for inducing PTPR $\square\square$ expression, down regulation of ERK phosphorylation and inducing apoptosis (Figure 6E), demonstrating that it is the sublocalization not the tether what makes the difference.

.....Other explanations are possible. Natively targeted HRAS17N is a potent dominant negative signaling element in the RAS/MAPK pathway because it is capable of sequestering GEFs. Might KDELR-HRAS V12 sequester some element of the signaling pathway that does not represent physiologic signaling? Indeed the data presented suggest that KDELR-HRAS V12 binds preferentially to RALGDS.

To the best of our knowledge, it has never been reported that RASV12 exerts some dominant inhibitory effect by unproductively sequestering some effector. But if so, why would it do such a thing exclusively at the Golgi and not at the plasma-membrane, for example?. Against this notion, in Fig 10 we now present new data indicating that in the absence of p53 KDELR-RAS can induce ERK activation in cells and melanomas in fish, behaving as expected.

Most of the data presented are results of overexpression of signaling molecules....

Strongly disagree. The aim of this study is to unravel RAS functions at the GC. To achieve

this we have looked at the activation of *endogenous* GC RAS by apoptogenic stimuli such as TGF β . And we have looked at the effects of sending *endogenous*, activated RAS to the GC via pharmacological means such as palmostatin B (and now 2-BP at your suggestion) or temperature shifts. These, absolutely physiological approaches, have indeed been complemented by the overexpression of signaling molecules in order to decipher the underlying molecular mechanisms.

....the only data to suggest that a similar phenomenon operates for endogenous RAS are shown in Fig. 3b where overexpression of an artificially Golgi targeted GEF downregulates ERK signaling in response to EGF or overexpression of HRAS12V. The physiologic correlate would be RASGRP1 signaling since it is the RAS GEF known to function at the Golgi.

Disagree. As mentioned above, the aim of this study is to unravel RAS functions at the GC. Overexpressed KDEL-CDC25 or RASGRP1 were utilized solely as tools to activate the endogenous RAS GC pool. We never intended to dissect the intimacies of GC RAS activation by upstream stimuli and GEFs.

But RASGRP1 potently stimulates ERK signaling (see for example PMID: 17283063). How do the authors reconcile this contradiction?

No contradiction, just cellular context-derived variability. The reviewer is absolutely right in that RASGRP1 activates ERK in some cell types, but this is not always the case. For example, in intestinal crypts RASGRP-1 antagonizes EGF-induced ERK activation and proliferative signaling, and RASGRP1 depletion results in exacerbated ERK activation and cellular proliferation (PMID: 26005835). In the same vein, while RASGRP1 induces proliferation in some cell types (PMID:12845332), it induces apoptosis in others (PMID: 14532295); both ERK-dependently (PMID:24515435) and ERK-independently (PMID: 14970203). Since RASGRP1 not only activates RAS at the GC but also at the plasma-membrane (PMID: 12845332), it is likely that an exquisite regulation of this GEF, in a cell and stimulus-dependent manner, orchestrates these antagonistic roles at the different sublocalizations where this GEF is active. In our case, we hypothesize that the balance is tilted towards RASGRP1 activating GC-RAS resulting in ERK down-regulation and apoptosis as opposed to plasma-membrane-RAS activation; ERK activation and proliferation. Certainly a scenario worthwhile investigating, though not in the context of the present study.

Have the authors considered examining RASGRP1 deficient cells?

RASGRP1 is mainly expressed in haemopoietic lineages (eg Jurkat cells). As it can be seen in the WB in Fig 4B, RASGRP1 expression levels are minimal, if any, in the cell lines under scrutiny in this study.

Is GTP loading of HA-KDELr-HRAS in response to TGF- β (Fig. 1c) dependent on RASGRP1?

In spite of minimal RASGRP1 expression levels in MCF-7 cells (Fig 4B), we have attempted to down-regulate them even further using siRNAs. As it can be seen in Figs 4C and D, this does not affect TGF β -induced RAS activation and apoptosis. So we must conclude that other GEFs are mediating in TGF β -induced effects. Maybe some other RASGRP family member?.

Does the effect of PalmB on selected tumor cell lines (Fig. 2d,f) depend on RASGRP1?.

As shown in Fig 4B, RASGRP1 expression levels are minimal in these cell lines. However,

we have used siRNAs as in MCF-7 cells and, as could be expected from the expression levels, found no differences. We have not included this negative data.

The SCG10-HRAS12V control shown in Supplemental Fig. 2 is compelling and underutilized. It would be informative to employ this construct in virtually all of the experiments, particularly the zebrafish studies shown in Fig. 6 and 7.

Agree. We have now used SCG10-RAS in Fig 5C to show that it does not antagonize ERK activation; in Fig 5E to show that it does not induce apoptosis. So we can conclude that cis- and trans-GC RAS do not behave equally in mammalian cells. We have also generated a transgenic fish, where we show that SCG10-RAS does not induce tumors (Supp Fig 7). It appears that in zebrafish GC-RAS, irrespective of cis- or trans, cannot elicit transforming signals. However, the new data that we have incorporated about p53, indicates that in its absence GC-RAS signaling turns tumourigenic.

How do the authors explain the paradoxical behavior of HeLa cells when KDELR-HRAS12V is overexpressed? Does this mean that the wiring of KDELR-HRAS12V as a negative regulator of ERK signaling and a positive regulator of apoptosis is cell type specific? How so?

As we show in Figure 6b, HeLa cells, do not express PTPRk. So as the reviewer states, the wiring of KDELR-HRAS12V as a negative regulator of ERK signaling and a positive regulator of apoptosis would be context specific depending on whether PTPRk is expressed or not. This notion gains further strength in light of our new data on p53. In Fig 6, we now show that in the absence of p53 KDELR-HRAS12V no longer induces PTPRk expression; no longer inhibits ERK activation; and does not induce apoptosis. In fact, in HeLa cells p53 is inactivated by HPV E6 protein, something that would explain the absence of PTPRk and the high ERK activation levels.

In Fig. 6c, why are the fish expressing KDELR-HRAS12V from a mitfa promoter more pigmented relative to the eGFP control, albeit in the absence of tumor? Does this not indicate proliferation rather than apoptosis of melanocytes?

The eGFP control animal and KDELR-HRAS12V animal were of different ages and the images taken under different lighting, hence the striping patterns somewhat differed, although neither animal displayed a disorganized striping pattern. The experiment has been repeated with age matched animals imaged under identical conditions and the pigmentation patterns are virtually indistinguishable.

Other specific comments:

It is not clear how apoptosis was scored. Results are reported as “fold change.” What does this mean? Was it the percentage of cells above some fluorescent annexin V gate in cytofluorometric analysis? How was that gate established?

When performing the Guava Nexin (EMD Millipore Guava Technologies) assay, samples were gated with X and Y intercepts between 10 (10e1) and 30 (10e3) on a log-fold scale at apparent breaks in cell populations, as illustrated in Supplemental Figure 2. Once gated, cells within the lower left quadrant were not labelled with either marker, therefore are not undergoing detectable apoptosis, while cells within the lower right quadrant were positive for Annexin-V-PE and negative for 7-AAD, marking them as early apoptotic cells. Those in the upper right quadrant were positive for Annexin-V-PE and negative for 7-AAD, indicating late

apoptosis. Very few cells were in the upper left quadrant and were not positive for the early apoptotic marker Annexin-V-PE, so were not considered. Three experiments were independently plated, with three replicate wells of each cell type per experiment. Apoptosis results are represented as the fold change of the Caspase 3/7 activity or Annexin V levels, relative to the untransfected or untreated control cells.

We are asked to make a leap of faith that Palm B or exposure to 21°C mislocalizes RAS to the Golgi in a number of tumor cell lines (Fig. 2d,e). This must be shown experimentally.

No leap of faith being asked because this effect had been published (PMID: 20418879). However, the reviewer is correct in that this control must be included. In Supp Fig 3 we now incorporate immunofluorescences for A375 and T24 cells showing that palmostatin B and 21°C treatments result in a profuse RAS accumulation at the Golgi complex.

Also, it is paradoxical that Palm B, which putatively inhibits depalmitoylation, promotes accumulation of HRAS on the Golgi. Also, Palm B has many off-target effects (PMID: 26701913). In contrast, 2-bromopalmitate (2-BP), which inhibits protein palmitoylation, allows NRAS and HRAS to accumulate on the Golgi for reasons that are easy to understand. What are the effects of 2-BP in this system?

Agree. This is a nice experiment and we have included this data in Supp Fig 4. 2-BP behaves as palm B inducing apoptosis in those cell lines expressing oncogenic H- and NRAS.

The blot of melanoma cells in Fig. 2e suggests that an HRAS-specific antibody was utilized. This is not likely the case since melanoma cells express relatively little HRAS and HRAS, in contrast to NRAS, has not been reported in cell supernatants (see for example PMID 27502489).

These are SKMEL2 cells that, in addition of harboring oncogenic NRAS, they express quite high levels of HRAS. HRAS and KRAS expression levels are highly variable in melanoma cell lines.

Also the arrows referred to in the legend are not shown.

Agree. Included

None of the immunoblots shown indicate MW. Whereas this is okay for routine demonstration of proteins like MEK, ERK and Tubulin, where fusion versus endogenous proteins come into play (e.g Fig. 1, 3c,d, 6a) MW markers would be helpful.

Agree. They have been included.

When results are shown as immunoblots the authors should indicate of how many independent experiments are the blots shown representative.

Agree. This information has been included in the M&M section. All experiments were repeated at least three times.

For many of the experiments shown (e.g. Fig. 2a, b, c, 3e, 5b) the relative expression level of the expressed proteins is critical to the interpretation of the results and must be shown.

Agree. They have been included.

In Fig. 7a is there a way to stratify the melanoma data by genotype? It would be informative to parse the expected 20% that are NRAS mutant from the majority that are BRAF mutant.

Unfortunately not. The raw data does not give figures for the different genotypes.

Reviewer #1, Expertise: melanoma zebrafish modelling (Remarks to the Author):

This is a much improved manuscript and the authors have made good efforts to address several of the key issues. Some minor questions that can be easily addressed:

1) In Figure 9c, I am not sure what the bottom fish is showing since it is not labelled as the ones above are. Also, in the middle fish is that supposed to be $ptprk^{+/+}$ not $-/-$?

2) In figure 10e, what are the actual numbers associated with these different groups? Do you have either a curve or an endpoint analysis?

Reviewer #2, Expertise: Ras and intracellular membranes

compartmentalization

(Remarks to the Author):

The authors have adequately addressed my critiques and those of another reviewer.

REVIEWERS REBBUTAL

1) In Figure 9c, I am not sure what the bottom fish is showing since it is not labelled as the ones above are. Also, in the middle fish is that supposed to be *ptprk*^{+/+} not *-/-*?

The information queried is correct as provided in the figure. As indicated in the figure legend [(C) View of naevi (2/28; arrowhead, middle panel) and melanoma (2/28; arrowhead bottom panel) by 52 weeks post-fertilisation in *ptprk* nullizygous zebrafish expressing *KDELr-HV12* as a transgene.] the arrow in the bottom panel points to a protruding eye with a retroorbital tumour. The arrow in the middle panel indicates a naevus in a *ptprk*^{-/-} animal.

2) In figure 10e, what are the actual numbers associated with these different groups? Do you have either a curve or an endpoint analysis?

This info is provided in Supplemental Figure 9 as a Kaplan-Meyer plot